# Extensive DNA methylome rearrangement during early lamprey embryogenesis

Allegra Angeloni[1,2], Skye Fissette [3], Deniz Kaya [4], Jillian M. Hammond [5,6], Hasindu Gamaarachchi [5,6,7], Ira W. Deveson [5,6,8], Robert J. Klose [4], Weiming Li [3], Xiaotian Zhang [9,11] & Ozren Bogdanovic [1,2,10]

DNA methylation (5mC) is a repressive gene regulatory mark widespread in vertebrate genomes, yet the developmental dynamics in which 5mC patterns are established vary across species. While mammals undergo two rounds of global 5mC erasure, teleosts, for example, exhibit localized maternal-to-paternal 5mC remodeling. Here, we studied 5mC dynamics during the embryonic development of sea lamprey, a jawless vertebrate which occupies a critical phylogenetic position as the sister group of the jawed vertebrates. We employed 5mC quantification in lamprey embryos and tissues, and discovered large-scale maternal-to-paternal epigenome remodeling that affects ~30% of the embryonic genome and is predominantly associated with partially methylated domains. We further demonstrate that sequences eliminated during programmed genome rearrangement (PGR), are hypermethylated in sperm prior to the onset of PGR. Our study thus unveils important insights into the evolutionary origins of vertebrate 5mC reprogramming, and how this process might participate in diverse developmental strategies.

DNA methylation (5-methylcytosine - 5mC) is a chemical modification to the DNA that represents one of the most pervasive gene regulatory marks in vertebrates[1–3]. 5mC is predominantly found within the cytosine-guanine dinucleotide context (mCG), occurring at ~80% of all CpG sites in vertebrate genomes[4–6]. mCG is associated with long-term silencing processes including somatic silencing of germline genes and silencing of repetitive DNA elements, as well as X-chromosome inactivation and genomic imprinting in mammals[7–10]. mCG is deposited by de novo DNA methyltransferases 3 A/B (DNMT3A/B) and maintained following DNA replication by DNA methyltransferase 1 (DNMT1)[11–13]. DNA demethylation can occur passively via replication-coupled dilution[14,15], or actively via methylcytosine dioxygenase Ten-Eleven Translocation (TET) enzymes[16].

Mammalian development is characterized by two waves of global mCG erasure occurring in the preimplantation embryo and in the developing germline, followed by re-establishment of cell-type specific mCG[17–24]. However, it appears that such mCG remodeling processes are not evolutionarily conserved. For example, the zebrafish genome does not undergo global DNA demethylation[10,25–27]. Instead, the maternal methylome is remodeled to match the hypermethylated paternal genome prior to the onset of zygotic genome activation (ZGA)[28]. Thus, mCG patterns in the early embryo closely resemble those of sperm. Moreover, we have recently identified maternal-to-paternal DNA methylome remodeling in medaka embryos, revealing evolutionary conservation of developmental epigenome dynamics in distantly related teleost species[29].

[1]Garvan Institute of Medical Research, Sydney, NSW, Australia. [2]School of Biotechnology and Biomolecular Sciences, University of New South Wales, Sydney, NSW, Australia. [3]Department of Fisheries and Wildlife, Michigan State University, East Lansing, MI, USA. [4]Department of Biochemistry, University of Oxford, Oxford, UK. [5]Genomics Pillar, Garvan Institute of Medical Research, Sydney, NSW, Australia. [6]Centre for Population Genomics, Garvan Institute of Medical Research and Murdoch Children's Research Institute, Darlinghurst, NSW, Australia. [7]School of Computer Science and Engineering, University of New South Wales, Sydney, NSW, Australia. [8]Faculty of Medicine, University of New South Wales, Sydney, NSW, Australia. [9]Center for Epigenetics, Van Andel Research Institute, Grand Rapids, USA. [10]Centro Andaluz de Biología del Desarrollo, CSIC-Universidad Pablo de Olavide-Junta de Andalucía, Seville, Spain. [11]Present address: University of Texas Health Science Center, Houston, TX, USA. ✉e-mail: o.bogdanovic@gmail.com

Unlike hypermethylated vertebrate genomes, genomes of non-vertebrate lineages typically display mosaic mCG patterns, characterized by regions of heavily methylated DNA interspersed with fully unmethylated domains[3,30,31]. To date, it remains elusive how and why vertebrate genomes transitioned to hypermethylation as a default state. The sea lamprey *Petromyzon marinus* is an extant jawless fish that can serve as a valuable model organism for understanding mCG evolution in metazoans. Lampreys are, together with hagfish, the sister group of jawed vertebrates (sharks to mammals), and the only living representatives of jawless vertebrates. Global mCG has been profiled in lamprey heart, brain, muscle and sperm, and was found to be highly heterogeneous at the majority of CpG sites, described as an intermediate between mosaic invertebrate mCG and vertebrate hypermethylation[32,33]. In addition, the lamprey genome contains a conserved mCG toolkit, including orthologues of DNMT1, DNMT3A, DNMT3B, DNMT3L, UHRF1, and TET proteins. Notably, lampreys as well as hagfish, only share one ancestral whole-genome duplication (WGD) round with jawed vertebrates, which makes them unique in terms of this major genomic event when compared to both invertebrates and vertebrates[34–36]. Moreover, lampreys undergo a peculiar biological phenomenon termed programmed genome rearrangement (PGR), in which genomic DNA present in the germline is physically eliminated in somatic lineages during early embryogenesis, and is effectively entirely removed from the genome three days post fertilization[37–41]. Homologs of genes in eliminated DNA sequences are enriched in functions related to germline development, and PGR is thought to prevent misexpression of genes with deleterious potential in somatic cells[42–44].

Here we sought to investigate whether the mCG configuration observed in the lamprey genome is compatible with embryonic epigenome remodeling processes, and whether sequences eliminated during PGR might be characterized by distinct 5mC patterning. To that end, we produced high resolution epigenome maps of sea lamprey development, employing whole-genome bisulfite sequencing (WGBS), biochemical identification of non-methylated DNA (BioCAP)[45] and

Nanopore sequencing[46] of germline, embryonic, and adult somatic tissues. Our results demonstrate that lampreys undergo large-scale maternal-to-paternal mCG remodeling. Unlike teleosts, however, where 5mC dynamics are localized to a relatively small number of defined loci, we discover developmental 5mC reprogramming occurring over partially methylated domains (PMDs) covering 29% of the entire genome, as well as at discrete gene regulatory elements (2% of the genome). Furthermore, we show that prior to the onset of PGR, regions eliminated during early embryogenesis are pre-targeted by DNA methylation in sperm. Our results shed new light on mCG dynamics in early vertebrate lineages and provide important insights into the origins and evolution of vertebrate mCG reprogramming.

## Results

### Disordered DNA methylation levels in lamprey tissues

To investigate the extent of mCG reprogramming in the lamprey genome, we first questioned whether there are any observable mCG changes between diverse embryonic and adult lamprey tissues. We performed WGBS on seven lamprey samples comprising germline (sperm and egg), early embryonic (day 1 and day 2) and adult somatic tissues (brain, muscle, and peripheral blood mononuclear cells—PBMCs) in biological replicates (Supplementary Fig. S1A, B, Supplementary Table S1). The embryonic stages correspond to 64 cell (day 1) and pre-ZGA blastula (day 2), with ZGA occurring ~2.5–3 days post fertilization[41]. Initial analysis of the generated data revealed that all lamprey tissues display genomic mCG levels at an average of ~29–40% at the majority of CpG dyads, in clear contrast to the methylomes of both divergent and closely related vertebrate and invertebrate species, in which most CpG sites display high or low mCG (Fig. 1A–C, Supplementary Fig. S1C)[3,30]. These results are suggestive of considerable variation in mCG at the cellular level and are in close agreement with previously described adult lamprey datasets[32,33]. As other vertebrates, lamprey exhibits depleted (<1%) mCG in the mitochondrial genome (Fig. 1B) and reduced non-CpG (CpH) genomic 5mC (Fig. 1D). The exception to these trends is

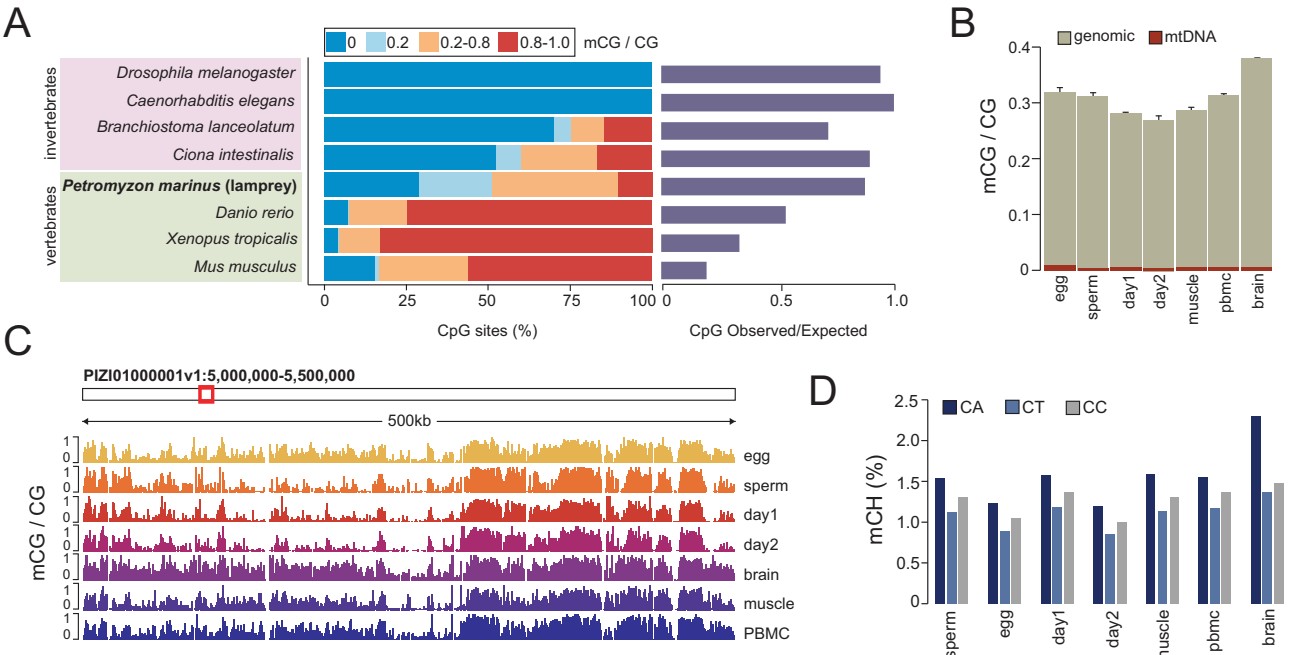

**Fig. 1 | Global DNA methylation levels in seven lamprey tissues. A** Per CpG DNA methylation levels (mCG) and CpG observed/expected content for eight metazoan genomes. Global mCG levels obtained from publicly available WGBS datasets[6,107,108]. **B** Genomic and mitochondrial mean mCG levels (*n* = 2 biological replicates). Error bars represent standard error of the mean (SEM). **C** Browser track depicting mCG profiles. mCG values represented in this figure correspond to merged WGBS replicates. **D** Total genomic mCH levels (%).

notable mCA enrichment (2.3%) in the brain, which was previously described as a deeply conserved epigenetic feature of the vertebrate neural system[33,47]. Altogether, the lamprey genome displays an intermediate DNA methylome state, in line with its phylogenetic position.

## Embryonic reprogramming of partially methylated domains

To better understand the origin and extent of intermediate DNA methylation in the sea lamprey genome, we partitioned the WGBS datasets into genomic blocks displaying highly disordered mCG levels (partially methylated domains, PMDs), and genomic regions located outside PMDs (non-PMDs) (Fig. 2A, Supplementary Data S1–14)[4,48]. We further divided the non-PMD regions of the genome into (i) unmethylated regions (UMRs) (ii) lowly methylated regions (LMRs), and (iii) hypermethylated regions (hyperMRs) (Supplementary Fig. S2A, B, Supplementary Data S8-14). It is worth pointing out that while PMDs in heavily methylated (>80% mCG) mammalian genomes typically display reduced 5mC levels compared to the global mean[4], PMDs in lamprey had heterogenous, but increased 5mC compared to non-PMD regions that generally showed low levels of 5mC (Fig. 2A), with the exception of hyperMRs (Supplementary Fig. S2B).To interpret similarities between PMDs in each sample, we compared the genomic locations of PMDs between tissues and found an increased association between PMDs in; (i) sperm, day 1 and day 2 methylomes, and in; (ii) egg, brain, muscle and PBMC methylomes (Fig. 2B). Overall, we found that 65% of the genome is classified as PMD in sperm, day 1 and day 2 tissues, in comparison to egg, brain, muscle and PBMC, where PMDs covered 46% of the genome (Supplementary Fig. S2A). We next quantified the percentage of the genome that transitions from PMD to non-PMD states and vice versa and found that on average 29% of the genome displays an altered PMD state in egg/sperm, egg/day 1 and egg/day 2 pairwise comparisons (Fig. 2C). Most of these transitions were characterized by PMD gain on the maternal genomic contribution prior to ZGA, to match the paternal methylome pattern (Fig. 2D, E). Specifically, 25% of the genome displayed PMD gain (UMR to PMD – 16%, hyperMR to PMD – 7%, and LMR to PMD – 2%), on the maternal genome, whereas 4% of the genome was characterized by maternal PMD to UMR transition (Fig. 2E).

As the regions undergoing reprogramming are partially methylated in either egg or sperm, we next took advantage of our single-molecule resolution data to study how this 5mC heterogeneity is generated at the individual read level, with the aim of resolving how partial 5mC states are established and lost during early pre-ZGA embryogenesis. To achieve this, we calculated the proportion of discordant reads (PDR – proportion of reads containing both methylated and unmethylated CpG sites)[49–51] within PMDs and plotted these values against PMD mCG levels (Fig. 2F). The maternal PMD gain (UMR to PMD) transition was characterized by an increase in both mCG levels and PDR, in line with unmethylated DNA gaining partial methylation, whereas the LMR and hyperMR to PMD groups displayed a broader range of mCG levels and decreased PDR in sperm, day 1 and day 2 samples (Fig. 2F). The second group, comprising PMDs present in egg, exhibited higher read discordancy and higher mCG levels in egg samples when compared to sperm and embryonic tissues, where these regions are classified as UMRs, and read discordancy and mCG levels were notably lower (Fig. 2F). Based on these results we conclude that maternal-to-paternal PMD reprogramming can involve either an increase (UMR to PMD, LMR to PMD) or a decrease (hyperMR to PMD) in mCG levels, whereas maternal PMD loss (PMD to UMR) is always associated with mCG loss. Moreover, our PDR analyses demonstrate that lamprey PMDs consist of a blend of discordant (partially methylated) and concordant (fully methylated or unmethylated) DNA molecules. Finally, we also observed minor PMD transitions between sperm/day 1 and sperm/day 2; however, those represented a relatively lower proportion of the entire genome and were generally shorter than

PMDs differentially identified between egg and sperm (Fig. 2C, Supplementary Fig. S2C). Overall, distinct genomic features such as UMRs, hyperMRs and LMRs are reprogrammed to PMDs on the maternal genomic contribution, prior to ZGA, with a smaller fraction of UMRs displaying PMD loss (Fig. 2F). These changes affect at least 29% of the genome, indicative of a major DNA methylome reconfiguration event.

## Differential DNA methylation at gene regulatory regions

Following observations that the lamprey epigenome could be partitioned into two major groups according to global tissue mCG patterning, we next wanted to explore whether similar distinctions could be observed at discrete regulatory regions. We focused this analysis on CpG islands (CGIs), also called non-methylated islands (NMIs), which are hypomethylated DNA sequences coinciding with vertebrate gene regulatory elements[52–55]. We experimentally identified NMIs in six lamprey tissues (Supplementary Table S2) (sperm, day 1, day 2, brain, muscle, PBMCs) using BioCAP, a biochemical method based on protein affinity pulldown of unmethylated CpG-rich DNA[45]. As with the DNA methylome, we found that the NMI signal clustered into two major groups when compared between tissues: (i) sperm, day 1, day 2, and (ii) brain, muscle, PBMC (Fig. 3A). This association was confirmed by Principal Component Analysis (PCA), (Supplementary Fig. S3A) and by k-means clustering of mCG levels at NMIs merged from all examined tissues (Fig. 3B, Supplementary Fig. S3B). We next identified a core set of NMIs present in all tissues ($n = 62,332$) (Fig. 3C, D, Supplementary Data S15), which are distinct from adult tissue-specific, or embryonic NMIs. We found DNA hypomethylation, increased CpG density and GC content, and association with accessible chromatin profiled in dorsal neural tube and whole heads[56], at core NMIs, thus confirming that the chromatin and sequence features of lamprey NMIs resemble canonical vertebrate CGIs (Fig. 3C, Supplementary Fig. S3C–E). We next performed motif calling on core NMIs and found enrichment for transcription factor binding sites associated with ubiquitously active CGI-like promoters in both vertebrate[57,58] and invertebrate[59] genomes, including the methyl-sensitive transcription factor E2F, the enhancer box (E-box) regulatory motif and the transcription factor nuclear respiratory factor (NRF) (Fig. 3E). Furthermore, we found that NMIs were associated with ~20–25% of all transcription start sites (TSS), underscoring their potential for gene regulatory function[53] (Fig. 3F, Supplementary Fig. S3F). Finally, we found a greater enrichment of brain/muscle/PBMC NMIs ($n = 24,381$) in genic regions compared to core NMIs and sperm/day 1/day 2 NMIs ($n = 9,895$), suggestive of differential mCG usage at NMIs in developmental and tissue-specific gene regulatory processes (Fig. 3G).

To obtain a more comprehensive view of developmental mCG dynamics in the lamprey, we performed pairwise comparisons of tissue- and stage-specific 5mC by identifying differentially methylated regions (DMRs) ($\Delta mCG > 0.2$, $p$ value < 0.05). This approach resulted in the identification of ~112,000 genomic regions displaying localized changes in methylation state (Fig. 3H, Supplementary Data S16). To determine tissue-specificity of DMRs, we assessed genomic co-localization of all pairwise DMRs (Fig. 3I). Overall, we found the greatest number of overlaps between egg/sperm, egg/day 1 and egg/day 2 DMRs, suggestive of maternal-to-paternal DMR reprogramming, as previously described in zebrafish and medaka[10,26–29]. Next, we merged all DMRs into a single dataset and performed clustering of mCG levels (Fig. 3J). mCG levels at DMRs grouped into two major tissue categories: (i) sperm, day 1 and day 2; and (ii) egg, brain, muscle and PBMC, in line with previous analyses. Notably these two major DMR groups were characterized by distinct features, including differences in mCG levels, genomic localization, CpG density distribution, PMD content, and BioCAP signal (Supplementary Fig. S4A–F), indicative of different biological functions (see discussion). Altogether, we identified that DMRs and differentially enriched NMIs represent 4% of the entire genome. While mCG patterns at these sequences recapitulate

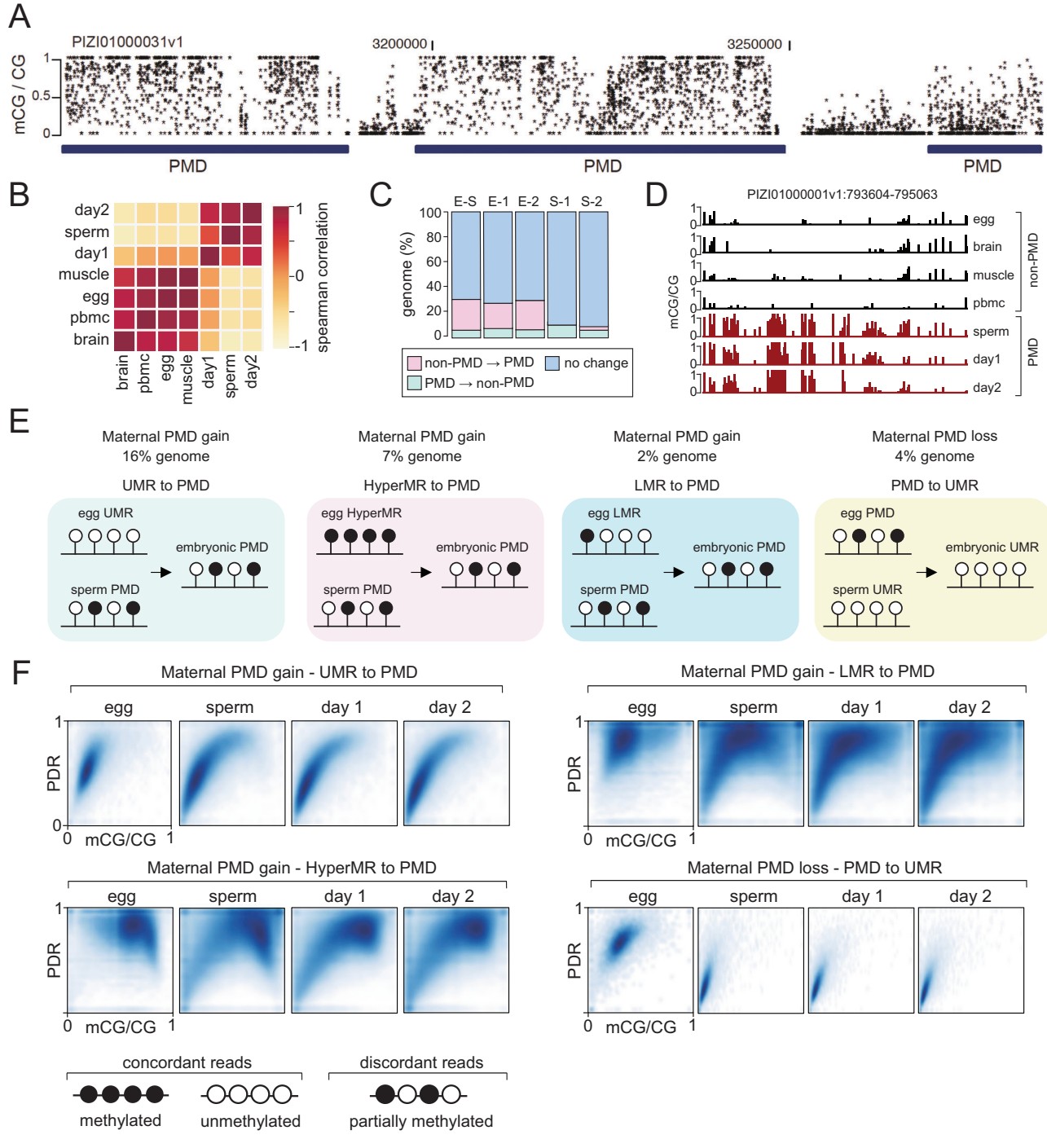

**Fig. 2 | Characterization of partially methylated domains in the lamprey genome. A** Browser track depicting per CpG mCG levels for PMDs (PMD partially methylated domain, dark blue) on scaffold PIZI01000031v1. All lamprey mCG values in this figure are from merged WGBS replicates. **B** Cluster heatmap of Spearman correlation of genomic PMD overlap. **C** Percentage of genome impacted by changes in PMD state in pairwise comparisons of samples (E egg, S sperm, 1 day 1, 2 day 2). **D** Browser track depicting mCG profiles of reprogrammed PMDs. **E** Schematic depicting major PMD reprogramming dynamics and the percentage of the genome associated with reprogramming events. UMR - unmethylated region, LMR - lowly methylated region, HyperMR- hypermethylated region. **F** Proportion of discordant reads (PDR) and mCG levels at reprogrammed PMDs. PDR and mCG levels are calculated from merged WGBS replicates for each tissue.

PMD dynamics, we found that 2% of the genome contains DMRs and differentially enriched NMIs located outside reprogrammed PMDs (Supplementary Fig. S4E). Finally, we wanted to assess whether DMRs and differentially enriched NMIs overlapping regulatory elements were linked to expression changes at corresponding genes. As 5mC-mediated CGI silencing is most commonly characterized by gain of

somatic mCG at gene promoters[10], we focused our analysis on promoters of protein-coding genes overlapping an NMI ($n = 5188$ genes), and assessed the correlation between NMI 5mC levels and transcription for an adult somatic tissue (brain) (Supplementary Fig. S4G)[60]. Overall, in line with canonical 5mC function, we identified a weak negative correlation between 5mC and transcription ($r = -0.23$). To

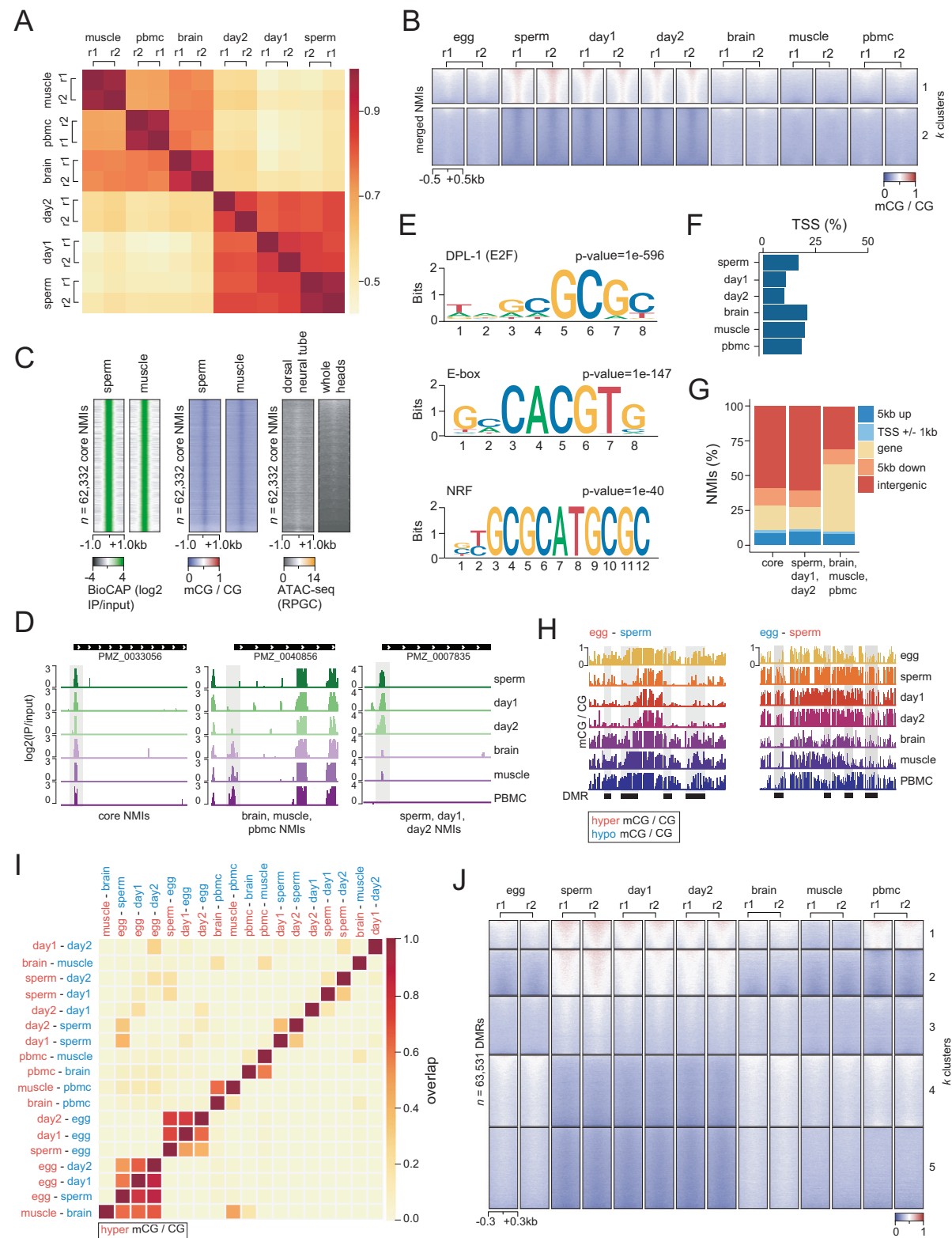

**Fig. 3 | Sequence and epigenetic features of differentially methylated regions.**
**A** Correlation of normalized read density at NMIs. r1 = BioCAP replicate 1, r2 = BioCAP replicate 2. **B** k-means clustering of mCG signal at merged NMIs ($k = 5$, 2 clusters shown). *Y* axis boxes refer to individual clusters. r1 = WGBS replicate 1, r2 = WGBS replicate 2. **C** BioCAP, mCG and ATAC-seq signal[56] at core NMIs (single replicate depicted for all heatmaps). **D** BioCAP signal at core and differential NMIs (highlighted in gray) overlapping transcription start sites (single replicate depicted). **E** Sequence motif enrichment at core NMIs. **F** Percentage of transcription start sites at protein-coding genes ($n = 20,895$) overlapped by NMI. **G** Percentage of differentially enriched NMIs at genomic features. **H** mCG profiles from merged WGBS replicates at DMRs (highlighted in gray). **I** Correlation of genomic DMR overlap. **J** k-means clustering ($k = 5$) of mCG signal at all DMRs. *Y* axis boxes refer to individual clusters. r1 = WGBS replicate 1, r2 = WGBS replicate 2.

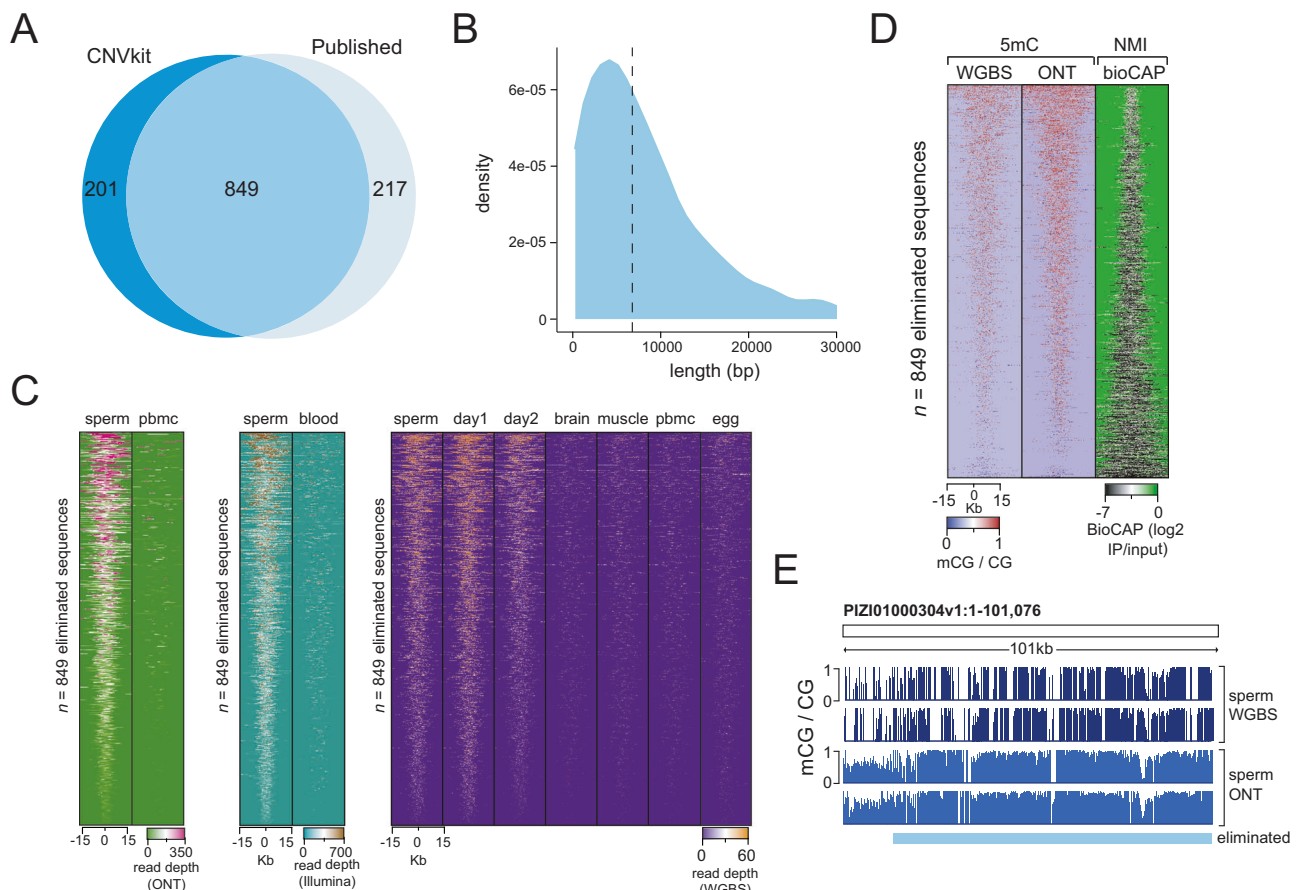

**Fig. 4 | DNA methylation at eliminated sequences. A** Overlap of eliminated sequences previously published[42] and eliminated sequences identified using CNVkit. **B** Distribution of lengths of eliminated sequences common to CNVkit and published data. Dotted line represents mean length. **C** Per-nucleotide read depth at eliminated sequences in whole-genome short-read[42], Oxford Nanopore Technologies (ONT), and bisulfite sequencing data (WGBS, single replicate depicted). **D** mCG, as measured by WGBS and ONT, and BioCAP signal at eliminated sequences in sperm samples (single replicate depicted). **E** Browser track depicting mCG profiles from sperm (in biological replicate) at an eliminated sequence.

study the dynamics of 5mC-mediated gene silencing during development, we next identified promoters of protein-coding genes overlapping both i) regions hypermethylated in egg/brain/muscle/PBMC compared to sperm/day 1/day 2, and (ii) NMIs enriched in sperm/day 1/day 2 compared to brain/muscle/PBMC (n = 13 genes) (Supplementary Fig. S4H). We then compared the expression profiles of these genes in diverse embryonic and adult somatic tissues[60]. Again, we observed an overall negative correlation between the presence of mCG and absence of NMI signal in adult tissues, and transcriptional activity, which was most notable in adult brain, kidney, and liver tissues (Supplementary Fig. S4H). Nevertheless, it is worth pointing out that relationships between promoter mCG and transcriptional silencing are complex[55,61,62] and that even in organisms with highly methylated genomes, like zebrafish and medaka[10,26,27,29], such clear anti-correlation can only be observed on a small number of genes. Altogether, our results indicate discrete mCG remodeling occurring not only within, but also alongside large-scale genomic transitions in PMD state during early development, affecting at least 2% of the genome.

### DNA eliminated during PGR is hypermethylated in sperm
PGR represents a unique biological mechanism for silencing potentially deleterious gene loci[37–42]. During PGR in lamprey, DNA sequences containing several hundreds of genes are physically eliminated from the genome during early embryogenesis, effectively producing a somatic genome that is a smaller, reproducible fraction of the germline genome. PGR eliminates potential for somatic misexpression of germline-specific genes and is thus analogous to epigenetic silencing mechanisms described in vertebrates, such as somatic gene silencing of cancer testis antigens[10,63–65]. As PGR is a crucial component of proper embryonic development in lamprey, we next interrogated the epigenetic landscape of sequences eliminated during PGR. We assembled whole-genome sequencing reads from blood and sperm[42] and called homozygous deletions in blood using CNVkit[66], identifying a total of 1050 DNA sequences that we classified as eliminated sequences. We found that ~81% of regions identified by our analysis overlapped previously published eliminated sequences[42] (Fig. 4A, B, Supplementary Fig. S5A). We then selected 849 regions (~9.3 Mb) overlapping both eliminated sequences identified by CNVkit and those previously published[42] as a stringent dataset for further genome-scale analyses (Fig. 4A, B, Supplementary Fig. S5A, Supplementary Data S17). These overlapping eliminated sequences contain 277 genes, the function of which have been described elsewhere[42–44] (Supplementary Fig. S5B, Supplementary Data S18). Also, in line with previous reports, we observed that a considerable percentage (~50%) of the eliminated fraction comprised diverse repetitive element classes, reflective of high genomic DNA repeat content (Supplementary Fig. S5C). We used whole-genome sequencing[42] and WGBS read coverage to confirm that the presence of eliminated sequences was restricted to the germline (sperm) (Fig. 4C, Supplementary Fig. S5D). We also identified increased read density at eliminated sequences in day 1 and day 2 WGBS datasets,

which is in agreement with previous findings that PGR occurs progressively during early development[37,40]. Finally, to confirm our findings using an orthogonal sequencing approach, we generated whole-genome Nanopore sequencing datasets for PBMCs and sperm in two biological replicates (Supplementary Table S3) and found increased read density at eliminated sequences in sperm, and read depletion in PBMCs (Fig. 4C, Supplementary Fig. S5E). Using our defined set of eliminated regions, we next set out to interrogate whether these genomic sequences exhibit a distinct mCG profile. Long-read sequencing is particularly useful for this analysis as eliminated sequences contain repetitive elements that may not be as well covered by short-read datasets (Supplementary Fig. S5C). Using sperm Nanopore sequencing data and sperm, day 1 and day 2 WGBS and BioCAP data, we identified DNA hypermethylation and depleted BioCAP signal at sequences eliminated during PGR (Fig. 4D, Supplementary Fig. S5F). We quantified mCG levels at these regions and found that the majority of CpG sites contained mCG >75%, indicating a clear exception to global heterogeneous mCG (Fig. 4E and Supplementary Fig. S5G, H). Nevertheless, we could not identify any notable sequence features that would distinguish these targeted sequences from the genomic background (Supplemental Fig. S5I), suggestive of sequence-independent mCG targeting. Although further research is necessary to clarify this, it is possible that mCG contributes to the selection or perhaps even germline protection of eliminated DNA, underscoring the significance of mCG dynamics in diverse developmental processes.

## Discussion

The goal of this study was to identify and explore the dynamics and conservation of developmental 5mC reprogramming in the basal vertebrate sea lamprey. Lamprey represents an important model organism for studies of developmental 5mC function as it displays a distinctive, highly heterogeneous DNA methylome and undergoes genome-wide structural rearrangement during early embryogenesis[32,37–42]. Our study establishes for the first time that ~30% of the lamprey genome undergoes extensive developmental mCG reprogramming. We demonstrate that mCG patterns in the early embryo closely resemble sperm, thereby reflecting a maternal-to-paternal epigenome transition event upon fertilization. Lamprey exhibits developmental mCG dynamics similar to zebrafish and medaka, thus providing valuable insights into the evolutionary origins of vertebrate mCG reprogramming[10,25–27,29] (Fig. 5). Given this similarity in early developmental epigenome dynamics to teleost fish, it will be of interest to investigate to what extent mCG contributes to the development of body plans[67] and phenotypic plasticity[68,69] in the agnathan lineage.

Evolutionary studies of developmental 5mC dynamics will be useful in clarifying why vertebrate genomes shifted to hypermethylation as a default state during the invertebrate-to-vertebrate transition[70,71]. One plausible hypothesis is that genomic hypermethylation was developed as a mechanism to fine tune the increase in regulatory complexity arising from WGD events in vertebrate lineages[7–10,72]. It is currently thought that the sea lamprey genome underwent only a single duplication event, while gnathostome (jawed vertebrate) genomes are shaped by two rounds of WGD[34–36]. It is therefore possible that major epigenetic reprogramming observed during lamprey development is an ancient mechanism developed to balance genomic hypermethylation, which could be linked to vertebrate genome expansion. While our results suggest that maternal-to-paternal epigenome reprogramming is likely conserved in anamniotes, we observed notable differences between lamprey and previously profiled teleost species in the extent of these processes. In zebrafish, much of developmental 5mC reprogramming is limited to discrete gene regulatory elements, such as promoter CGIs[10,26,27]. In the lamprey however, while we identify changes to 5mC levels at putative regulatory elements, we also describe vast changes in genomic 5mC

content between egg and sperm, with the sperm methylome being inherited in the early embryo. Approximately 29% of the lamprey genome transitions in PMD state post fertilization (Fig. 5), representing a large-scale developmental 5mC remodeling event in comparison to other anamniotes profiled to date. Moreover, we observed that maternal-to-paternal reprogramming of PMD states involves both developmental gain and loss of PMDs on the maternal allele, even though developmental PMD gain before the ZGA onset was the predominant transition, affecting ~25% of the genome.

It remains elusive exactly how mCG remodeling is achieved in lamprey. Based on mCG reprogramming dynamics identified in our study, it is not unlikely that lamprey utilize placeholder nucleosomes enriched in H2A.Z and H3K4me1 in a similar manner to zebrafish[28]. These placeholder nucleosomes are specific to the paternal germline in zebrafish and establish pre-ZGA chromatin in the early embryo. In zebrafish, placeholder nucleosomes are found at hypomethylated regulatory regions associated with developmental and housekeeping genes, where they deter DNMT activity while maintaining a transcriptionally quiescent state during cleavage stages[28]. Thus, it is plausible that NMIs and hypomethylated DMRs in sperm and embryonic tissues, represent sequences associated with a similar chromatin configuration in lamprey, which consequently display reduced mCG levels compared to tissues where such nucleosome positioning may not be present. In terms of DMRs that we find hypermethylated in lamprey sperm and embryonic tissues, a recent zebrafish study identified that DNA hypermethylation of CpG-rich enhancers in sperm and pre-ZGA embryos safeguards embryonic programs by preventing premature activation of transcriptional programs associated with adult tissues[67]. Depletion of *dnmt1* resulted in severe developmental defects and embryonic lethality that was linked to ectopic activation of these enhancers, emphasizing that inheritance and maintenance of paternal-like 5mC states plays a critical role in regulating developmental programs. It is important to note that in zebrafish, establishment of the embryonic epigenome is not dependent on the paternal genome; even in parthenogenetic embryos (maternal haploids that lack sperm DNA), a sperm-like chromatin configuration is still observed[27,28,67]. This suggests that in zebrafish, and probably also lamprey, the embryonic methylome is already established in transcriptionally quiescent sperm, potentially to ease the establishment of totipotency by only reprogramming one parental allele. In addition, H3K27me3 is a repressive chromatin modification deposited by Polycomb-group proteins, and is widely recognized to mediate silencing of developmental genes[73,74]. Some murine homologs of genes eliminated during PGR were found to be targets of Polycomb repressive complexes in mouse embryonic stem cells[42]. It will therefore be of particular importance to understand the genomic distribution and concentration of H3K27me3-marked regions in lamprey compared to vertebrate species that do not undergo PGR.

Our quantitative, base-resolution results of mCG targeting demonstrate that genomic regions eliminated during PGR are hypermethylated in sperm. This indicates that prior to fertilization and the onset of PGR, eliminated sequences are already epigenetically modified in the germline, which may facilitate accurate targeting and removal during PGR. Previous experiments performed using immunofluorescence assays suggested that mCG modification of targeted sequences could occur following their elimination and packaging into micronuclei[40]. It is important to note that the regions used in this study do not represent all sequences eliminated during PGR, due to technical challenges associated with the abundance of highly repetitive sequences in eliminated DNA. PGR is not a lamprey-specific biological phenomenon; it has been described in several protozoan, invertebrate and vertebrate species, albeit with differences in timing and molecular mechanisms[75,76]. A common theme in PGR across diverse species is that eliminated sequences are associated with heterochromatin, such as in zebra finch[77], the ciliated protozoan *Tetrahymena*[78,79], and in sciarid

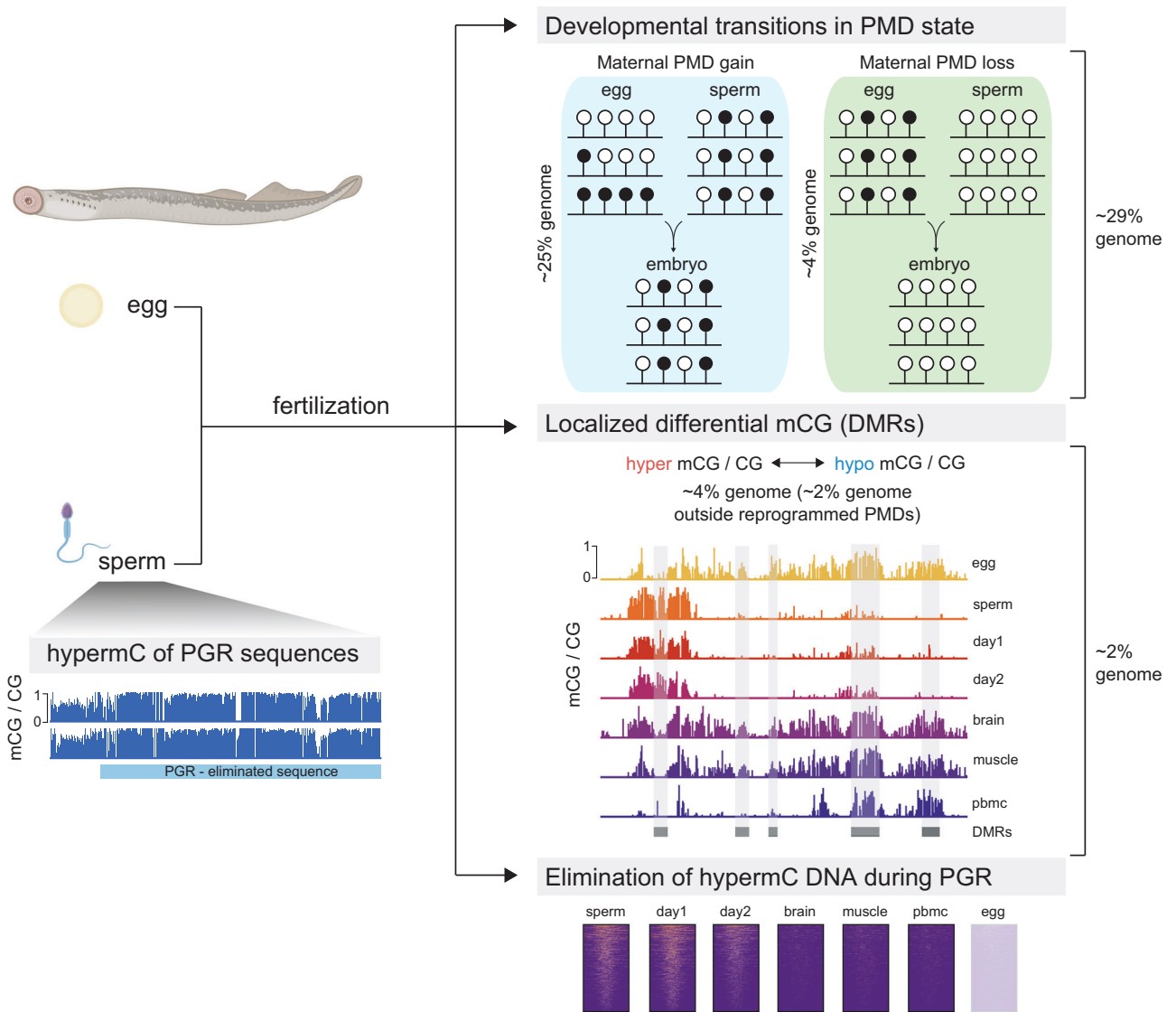

**Fig. 5 | Overview of developmental 5mC reprogramming in the sea lamprey.** Early embryogenesis in the sea lamprey is characterized by maternal-to-paternal mCG reprogramming that affects up to 31% of the embryonic genome. These epigenetic remodeling events occur both on a large-scale (PMDs, 29%) and at discrete gene regulatory elements (DMRs, NMIs – 2% of the genome, outside of PMDs). Reprogramming of PMD states is predominantly characterized by transitions from UMRs to PMDs and vice versa. Sequences that are eliminated during PGR in the early embryo are pre-targeted by DNA methylation in sperm. More research is needed to unequivocally determine whether the eliminated sequences are present or absent in the adult female germline. Created with BioRender.com.

flies[80,81]. Currently, genomic and epigenetic studies of PGR in lamprey have mostly been limited to analyses of the male germline. However, in some species, such as the zebra finch, the germline-restricted chromosome, which is found in both the male and the female germline, is eliminated from mature sperm[82,83]. In our study, we found that WGBS read coverage at eliminated sequences in the egg was comparable to that of somatic tissues, suggestive of some degree of DNA elimination. Previously, RNA-seq analyses of male and female lamprey gonad development have shown that eliminated genes are 36X more likely to be expressed in testes than in ovaries and that strikingly, ~70% of those genes were not expressed in differentiated ovarian samples[43]. This indeed suggests that the complement of eliminated genes might differ between the male and female germline and that programmed DNA loss from mature oocytes is indeed a possibility. Nevertheless, given the high content of repetitive DNA at those regions and the fact that our observations are based on WGBS data alone, at this point no conclusive statement can be made regarding the structure and content of PGR sequences in the female germline. Further studies, preferably

involving long-read DNA sequencing technologies, will be required to resolve this issue. Finally, an important question that remains open is how these regions remain refractory to elimination during germ cell development, which may be explained at least in part by the DNA hypermethylation observed in this study. Unlike mammals, fish do not undergo global mCG reprogramming in primordial germ cells, rather the paternal mCG configuration is retained[10,84]. If a similar mechanism would be observed in the lamprey, it would be plausible to suggest that DNA hypermethylation at eliminated sequences in sperm contributes to their retention in germline lineages. Further studies involving genomic localization of DNMTs during lamprey embryonic development, including co-factors that are necessary for DNMT targeting to defined genomic locations, will deepen our understanding of the reprogramming processes described in this study.

In summary, we have demonstrated that despite high levels of mCG heterogeneity, the sea lamprey undergoes extensive maternal-to-paternal developmental DNA methylome remodeling predominantly associated with partially methylated DNA states. Moreover, we have

shown that regions eliminated during PGR are marked by high mCG levels in sperm and pre-ZGA embryos. Altogether, this research demonstrates that the last common ancestor of vertebrates might have already presented extensive mCG reprogramming, and that this occurred before the second round of WGD that characterizes jawed vertebrates. Lampreys show lineage specific variations in the way in which they establish developmental mCG patterns, yet their epigenome configuration is suggestive of an early establishment of complex regulatory states that characterize the vertebrate lineage.

## Methods

### Lamprey procedures
All lamprey embryonic and adult material was collected at the Hammond Bay Biological Station (Michigan, USA). Embryos were obtained and grown as previously described[85]. All experimental procedures for culturing embryos were approved by Michigan State University Institutional Animal Care and Use Committee (AUF# 02/17-031-00). To produce sexually mature males and females for embryo fertilization, sea lamprey were transferred to the Ocqueoc River, Millersburg Michigan and held in cages (0.5 m³) to allow natural sexual maturation in a riverine environment. Sea lamprey were checked daily for sexual maturity; sexually mature individuals were identified by applying abdominal pressure and checking milt expression or for ovulated oocyte expression. Sexually mature males and female lamprey were returned to HBBS and held until use for culturing lamprey embryos.

### Genomic DNA extraction from lamprey germline, embryonic and somatic tissues
All lamprey samples were lysed in buffer containing 20 mM Tris, pH 8.0, 100 mM NaCl, 15 mM EDTA, 1% SDS and 0.5 mg/ml proteinase K for 3 h at 55 °C. Additionally, before DNA extraction, the egg samples were rinsed extensively under distilled water to eliminate debris. Lysis was followed by two phenol:chloroform:isoamyl alcohol (25:24:1) extractions and subsequent centrifugation (5 min at 17,949 × g). DNA was then precipitated by adding 0.2 volumes of 4 M ammonium acetate and three volumes of 96% ethanol. The reaction was left on ice for a minimum of 30 min. The DNA precipitate was centrifuged for 20 min at 4 °C (17,949 × g), and the pellet was washed with 500 µl of 70% ethanol and centrifuged for 5 min (17,949 × g) at room temperature. The pellet was then resuspended in 200 µl of TE buffer, and 1 µl of RNase A (20 µg/µl) was added. The reaction was left to proceed for 30 min at room temperature, after which time the DNA was precipitated with 0.1 volumes of 4 M ammonium acetate and 1 volume of isopropanol on ice for 2 h. The tubes were centrifuged for 30 min at 4 °C (17,949 × g), and the pellet was resuspended in TE buffer.

### WGBS library preparation
WGBS was performed as described previously[10]. Briefly, DNA was spiked with unmethylated lambda phage fragments (Promega) and sonicated to approximately 300 base pairs. Sonicated DNA was concentrated in a vacuum centrifuge to a final volume of 20 µL, required for bisulfite conversion with the EZ DNA Methylation-Gold Kit (Zymo Research). Bisulfite-converted DNA was then subjected to low input library preparation using the Accel-NGS Methyl-seq DNA Kit (Swift Biosciences). Briefly, the single-stranded, bisulfite-converted DNA was subjected to an adaptase reaction, followed by primer extension, adapter ligation (Methyl-seq Set A Indexing Kit-Swift Biosciences), and indexing PCR. Libraries were quantified using KAPA qPCR Library Quantification Kit (KAPA Biosystems) according to manufacturer instructions. Methylome libraries were sequenced on the Illumina HiSeqX platform (high-throughput mode, 150 bp, paired-end).

### WGBS read assembly
Read assembly was performed as described previously[10]. Briefly, sequenced reads in FASTQ format were trimmed using the fastp v0.12.5 tool (https://github.com/OpenGene/fastp) with the following settings: (fastp -i ${read_1} -I ${read_2} -o ${trimmed_read_1} -O ${trimmed_read_2} -f 10 -t 10 -F 10 -T 10). Trimmed reads were mapped (petMar3 genome reference, containing the lambda genome as chrL) using WALT with the following settings: -m 10 -t 24 -N 10000000 -L 2000 (https://github.com/smithlabcode/walt). The petMar3 germline reference genome is available from https://www.ncbi.nlm.nih.gov/datasets/genome/GCA_002833325.1. PCR and optical duplicates were removed using Picard tools v2.3.0 (https://broadinstitute.github.io/picard/). Genotype and methylation bias correction was performed using MethylDackel software (https://github.com/dpryan79/MethylDackel). The number of methylated and unmethylated calls at each genomic CpG position were determined using MethylDackel (MethylDackel extract genome_lambda.fa $input_bam –mergeContext –minOppositeDepth 10 –maxVariantFrac 0.5 $input_bam -@ 32 –OT 0,0,0,0 –OB 10,0,0,0). Bisulfite conversion efficiency was estimated from the lambda phage spike-ins. Bedgraphs were converted to bigWig format using the bedGraphToBigWig script from kentUtils (https://github.com/ENCODE-DCC/kentUtils).

### WGBS methylome PCA
PCA of mCG levels was calculated using methylKit PCASamples on two replicates of tissue methylomes using default parameters[86].

### Identification of partially methylated domains
Methylome replicates were merged and PMDs were called using MethylSeekR segmentPMDs function on scaffold PIZI01000001v1 using default parameters[48]. UMRs and LMRs were called using MethylSeekR segmentUMRsLMRs function. For UMRs and LMRs that partially overlap PMDs, intersecting bases are counted as PMDs for further analysis. hyperMRs are defined as regions that are not called as UMRs, LMRs or PMDs by MethylSeekR. Tissue PMDs were intersected using Intervene pairwise (–compute frac) and Spearman correlation was found using the Intervene web application (https://asntech.shinyapps.io/intervene/)[87]. mCG levels from WGBS methylomes at PMDs and non-PMDs were found using bedtools intersect function[88]. Reprogrammed PMDs were identified through pairwise intersections of PMD and non-PMD regions using bedtools intersect function. To identify the percentage of discordant reads (PDR) at egg non-PMDs/sperm PMDs, per-read mCG status was determined from WGBS datasets using the MethylDackel perRead function. Discordant reads with at least four CpG sites were intersected with egg non-PMDs/sperm PMDs using bedtools coverage function (-counts) and mCG status was found by intersecting WGBS datasets with egg non-PMDs/sperm PMDs using bedtools map function (-o mean -null na). Scatterplots of PDR and mCG were generated using the smoothScatter function in R (nrpoints = 0). PMD genome coverage is calculated relative to the size of the petMar3 reference genome.

### BioCAP library preparation
BioCAP was performed on genomic DNA extracted from sperm, day 1, day 2, brain, muscle and PBMC in replicate and with input controls using published methods[89]. Briefly, genomic DNA was sonicated to approximately 200 base pairs and incubated with biotinylated zinc-finger CxxC protein domain from human KDM2B. NMIs were eluted in buffer containing 700mM-1M NaCl, 0.1% Triton X-100, 20 mM HEPES pH 7.9 and 12.5% v/v glycerol, then DNA was purified using the Wizard SV Gel and PCR Clean-Up System (Promega) according to manufacturer instructions. Library preparation was performed on BioCAP samples using TruSeq ChIP Sample Preparation (Illumina) according to manufacturer instructions. Libraries were amplified using 18 PCR cycles and quantified using KAPA qPCR Library Quantification Kit (KAPA Biosystems) according to manufacturer instructions. Libraries were sequenced on the Illumina NovaSeq 6000 (150 bp, paired-end), generating 33M-307M read pairs per sample.

## BioCAP read assembly

BioCAP reads in FASTQ format were trimmed using trimmomatic (ILLUMINACLIP:TruSeq3-PE-2.fa:2:30:10 SLIDINGWINDOW:5:20 LEADING:3 TRAILING:3 MINLEN:25)[90] and aligned to the petMar3 reference genome using bowtie2 (-p 10 -N 1 –very-sensitive -X 2000 –no-mixed –no-discordant)[91]. The resulting alignments in BAM format were deduplicated using Picard MarkDuplicates (REMOVE_DUPLICATES=true). NMIs were identified using the macs2 callpeak function[92]. When replicate is not specified, dataset refers to NMIs common to both replicates. Bigwig (log2 IP/input) files were generated using deepTools bamCompare (–ignoreDuplicates –binSize 50 –centerReads –extendReads)[93].

## ATAC-seq read assembly

Publicly available ATAC-seq data in FASTQ format was downloaded[56]. ATAC-seq reads were trimmed using trimmomatic (ILLUMINACLIP:TruSeq3-PE-2.fa:2:30:10 SLIDINGWINDOW:5:20 LEADING:3 TRAILING:3 MINLEN:25) and aligned to the petMar3 reference genome using bowtie2 (-p 10 -N 1 –very-sensitive -X 2000 –no-mixed –no-discordant). To include only nucleosome-free read alignments, alignments exceeding 100 bp were removed. Aligned reads were deduplicated using Picard MarkDuplicates (REMOVE_DUPLICATES=true). Bigwig (RPGC) files were generated using deepTools bamCoverage (–normalizeUsing RPGC –numberOfProcessors 4 –ignoreDuplicates –binSize 50 –centerReads –extendReads –effectiveGenomeSize 1040808129).

## Epigenetic features of NMIs

Clustering and PCA of NMI read counts was performed using DiffBind dba.count and dba.plotPCA functions, respectively[94]. To calculate mCG clustering, all NMI datasets were concatenated and overlapping sequences were merged using bedtools merge function. Heatmaps of mCG signal at merged NMIs were generated using deepTools computeMatrix function (computeMatrix reference-point –referencePoint center –binSize 10 –afterRegionStartLength 500 –beforeRegionStartLength 500), with replacement of NaN values with mean mCG (0.33) after the matrix file was generated, followed by the plotHeatmap function (–k-means 5 –yMin 0 –yMax 1). NMIs present in every tissue were found by intersecting all NMI datasets using bedops intersect function[95]. Heatmaps of BioCAP signal at core NMIs were generated using deepTools computeMatrix function (computeMatrix reference-point –referencePoint center –binSize 10 –afterRegionStartLength 1000 –beforeRegionStartLength 1000 –missingDataAsZero), followed by the plotHeatmap function (–sortRegions no). Heatmaps of mCG signal at core NMIs were generated using deepTools computeMatrix function (computeMatrix reference-point –referencePoint center –binSize 10 –afterRegionStartLength 1000 –beforeRegionStartLength 1000), with replacement of NaN values with mean mCG for each tissue after the matrix file was generated, followed by the plotHeatmap function (–sortRegions no). Heatmaps of ATAC-seq signal at core NMIs were generated using deepTools computeMatrix function (computeMatrix reference-point –referencePoint center –binSize 10 –afterRegionStartLength 1000 –beforeRegionStartLength 1000 –missingDataAsZero), followed by the plotHeatmap function (–sortRegions no).

## Sequence motifs at core NMIs

Enriched sequence motifs at core NMIs were found using Homer findMotifsGenome.pl function (-size 200) with default parameters[96]. Sequence logos were visualized using the ggseqlogo package in R[97]. CpG observed/expected and GC content was calculated using bedtools getfasta and an in-house generated R script.

## Localization of NMIs at genomic features

NMIs differentially enriched in sperm/day 1/day 2 and brain/muscle/PBMC were found using DiffBind dba.analyze function (false discovery rate <0.05, fold change >2). petMar3 gene predictions were downloaded[42] and overlap with NMIs was identified using bedtools intersect function. Barplots were generated using ggplot2 geom_bar function in R[98]. Heatmaps of BioCAP signal at TSS were generated using deepTools computeMatrix function (computeMatrix reference-point –referencePoint center –binSize 10 –afterRegionStartLength 1000 –beforeRegionStartLength 1000 –missingDataAsZero), followed by the plotHeatmap function (–sortRegions no).

## Differentially methylated regions

DMRs were called from WGBS data (replicates merged) with the DSS software[99,100] with the following parameters: delta=0.2, p.threshold=0.05, minlen=100, minCG=10, dis.merge=100. DMRs were intersected using Intervene pairwise (–compute frac) and the heatmap was generated using the Intervene web application (https://asntech.shinyapps.io/intervene/). To calculate mCG clustering, all DMRs were concatenated and overlapping sequences were merged using bedtools merge function. Heatmaps of mCG signal at merged DMRs were generated using deepTools computeMatrix function (computeMatrix reference-point –referencePoint center –binSize 10 –afterRegionStartLength 300 –beforeRegionStartLength 300), with replacement of NaN values with mean mCG (0.33) after the matrix file was generated, followed by the plotHeatmap function (–k-means 5 –yMin 0 –yMax 1). mCG levels at DMRs were found using bedtools intersect function and histograms of mCG at DMRs were created using ggplot2 geom_histogram function. Heatmaps of BioCAP signal at DMRs were generated using deepTools computeMatrix function (computeMatrix reference-point –referencePoint center –binSize 10 –afterRegionStartLength 300 –beforeRegionStartLength 300 –missingDataAsZero), followed by the plotHeatmap function (–sortRegions no). CpG density was calculated using bedtools getfasta and an in-house generated R script.

## Localization of DMRs at genomic features

petMar3 gene predictions were downloaded[42] and overlap with DMRs was identified using bedtools intersect function. Barplots were generated using ggplot2 geom_bar function in R.

## Transcriptome analyses

RNA-seq reads in FASTQ format were downloaded from[60]. Reads were trimmed for Illumina sequencing adapters using trimmomatic and mapped with Kallisto[101] with the following settings: -l 300 -s 100 –single to the PMZ_v3.1.gff transcriptome reference. Mean 5mC from merged brain WGBS datasets was calculated at all promoters of protein-coding genes (TSS ±1 kb) containing an NMI directly overlapping the TSS using bedtools intersect function and bedtools map function (-o mean -null na). 5mC levels were plotted against adult brain transcription levels using ggplot2 geom_point function in R. Correlation was calculated using ggplot2 geom_smooth function (method=lm, se=FALSE) and cor() function in R.

## Nanopore library preparation and sequencing

Library preparation was performed using ONT ligation sequencing (SQK-LSK110). Samples were sequenced across two PromethION (FLO-PRO002) flow cells for 72 hours.

## Nanopore read assembly and mCG calling

Raw ONT sequencing data was converted to BLOW5 format[102] using slow5tools v0.3.0[103] then base-called using Guppy v5.0.13 (high-accuracy model). Resulting FASTQ files were aligned to the petMar3 reference genome using minimap2 v2.22[104]. 5mC profiling on CpG sites within petMar3 was performed using f5c v0.7[105] with BLOW5 data input. CpG methylation frequencies were determined using the methfreq tool in f5c.

## Identification of sequences eliminated during PGR

Publicly available blood and sperm whole-genome sequencing reads in FASTQ format were downloaded[42]. Reads were trimmed using trimmomatic and aligned to the petMar3 reference genome using bowtie2 (-p 10 -N 1 −very-sensitive -X 2000 −no-mixed −no-discordant). Read alignments in BAM format were input into CNVkit (cnvkit.py -batch -m wgs -f petMar3.fa) to identify sequences eliminated during PGR, with blood used as input and sperm used as the control sample (−normal)[66]. We defined homozygous deletions as regions with a copy number of 0 in blood. We intersected eliminated sequences identified by CNVkit with a set of previously published eliminated sequences[42] as a stringent dataset for further analysis. Sequences were validated using coverage metrics from short-read whole-genome sequencing, WGBS and Nanopore sequencing. Read depth was calculated using samtools depth function[106] and converted to bigWig format using the bedGraphToBigWig script from kentUtils. Heatmaps of read coverage at eliminated sequences were generated using deepTools computeMatrix function (computeMatrix reference-point −missingDataAsZero −binSize 10 −afterRegionStartLength 15000 −beforeRegionStartLength 15000), followed by the plotHeatmap function. petMar3 gene predictions were downloaded[42] and overlap with eliminated sequences was identified using bedtools intersect function.

## mCG at eliminated sequences

Heatmaps of mCG signal at eliminated sequences were generated using deepTools computeMatrix function (computeMatrix reference-point −binSize 10 −afterRegionStartLength 15000 −beforeRegionStartLength 15000), with replacement of NaN values with mean mCG for each sequencing technique after the matrix file was generated, followed by the plotHeatmap function. Heatmaps of BioCAP signal at eliminated sequences were generated using deepTools computeMatrix function (computeMatrix reference-point −binSize 10 −afterRegionStartLength 15000 −beforeRegionStartLength 15000 −missingDataAsZero), followed by the plotHeatmap function. mCG status at eliminated sequences was found by intersecting WGBS datasets with eliminated sequences using bedtools intersect function and violin plots were generated using ggplot2 geom_violin function in R. Repeat content was found by intersecting the petMar3 RepeatMasker track with eliminated sequences using bedtools intersect function. Dinucleotide frequency at eliminated sequences was calculated using the faCount script from kentUtils. Correlation between Nanopore and WGBS datasets was calculated and plotted using methylKit getCorrelation function (plot=TRUE) in R.

## Reporting summary

Further information on research design is available in the Nature Portfolio Reporting Summary linked to this article.

## Data availability

Raw and processed BioCAP-seq and WGBS data generated for this study are available from NCBI Gene Expression Omnibus under accession code GSE220553 [https://www.ncbi.nlm.nih.gov/geo/query/acc.cgi?acc=GSE220553]. Nanopore sequencing data generated for this study is available from NCBI under accession code PRJNA783432 [https://www.ncbi.nlm.nih.gov/bioproject/PRJNA783432]. ATAC-seq data used in this study is available from NCBI Gene Expression Omnibus under accession code GSE112072 [https://www.ncbi.nlm.nih.gov/geo/query/acc.cgi?acc=GSE112072]. Whole-genome sequencing data used in this study is available from NCBI Sequence Read Archive under accession codes SRR5535434 [https://www.ncbi.nlm.nih.gov/sra/?term=SRR5535434] and SRR5535435 [https://www.ncbi.nlm.nih.gov/sra/?term=SRR5535435]. RNA-seq data used in this study is available from NCBI under accession code PRJNA50489 [https://www.ncbi.nlm.nih.gov/bioproject/50489]. All sequencing data was aligned to the petMar3 reference germline genome available from NCBI under GenBank accession code GCA_002833325.1 [https://www.ncbi.nlm.nih.gov/datasets/genome/GCA_002833325.1]. Source data are provided as a Source Data files.

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

## Acknowledgements

The Australian Research Council (ARC) Discovery Project (DP190103852); Ramón y Cajal fellowship (RYC2020-028685-I) and the "Proyecto de Generación de Conocimiento 2021" project (PID2021-128358NA-I00) from the Spanish Ministry of Science and Innovation, as well as funding from CEX2020-00108-M Unidad de Excelencia María de Maeztu to OB supported this work. Figures were created with the help of BioRender.com software. The authors thank Alex de Mendoza for the critical reading of the manuscript.

## Author contributions

O.B. conceived the study. Lamprey embryo and adult tissue collection and DNA extraction were performed by O.B., X.Z., W.L. and S.F. Sequencing libraries were generated by A.A. CxxC protein domain from human KDM2B used for BioCAP was prepared by D.K. and R.J.K. Nanopore library preparation and sequencing were performed by J.M.H. Nanopore read assembly and 5mC calling was performed by H.G. and I.W.D. A.A. performed all other data analysis. O.B. participated in WGBS data analysis. A.A. and O.B. wrote the manuscript. All authors contributed to, read, and approved the final manuscript.

## Competing interests

IWD manages a fee-for-service sequencing facility at the Garvan Institute of Medical Research that is a customer of Oxford Nanopore Technologies but has no further financial relationship. HG and IWD have previously received travel and accommodation expenses to speak at Oxford Nanopore Technologies conferences. The remaining authors declare no competing interests.
