## [Peer Review File · Nature Communications]

Extensive DNA methylome rearrangement during early lamprey embryogenesisREVIEWER COMMENTS

Reviewer #1 (Remarks to the Author):

Extensive DNA methylome rearrangement during early lamprey embryogenesis

Overall, this is an interesting paper by Angeloni et al on changes in DNA methylome during early lamprey embryogenesis. The Bogdanovic lab has pioneered new approaches to studying changes in methylation patterns in vertebrates during early embryogenesis. Broadly, there is evidence of major change in global early developmental methylation from invertebrates to mammals, with invertebrates showing very low levels of methylation, while mammalian genomes are hypermethylated. Additionally, there are major differences across vertebrates in the re-programming of methylation marks in early embryogenesis with mammals exhibiting nearly complete erasure of the gametic methylation patterns while in zebrafish, the embryo inherits the methylation pattern found in the sperm. Lying at the base of the vertebrate evolutionary tree, sea lamprey are thus an interesting taxa in which to study global methylation across tissues and developmental stages, as well as the inheritance (or lack thereof) of methylation patterns following fertilization. Furthermore, since lamprey undergo a process termed programmed genome rearrangement (PGR), it is of interest to assess if the regions of the lamprey genome that are eliminated during PGR have differential methylation from the non-eliminated regions.

Angeloni et al., use whole genome bisulfite sequencing (WGBS) to assess global differences in methylation in somatic tissues - muscle, brain and PBMC's, as well as gametes (eggs and sperm) and embryos 1 and 2 dpf, since PGR is complete between days 2.5-3. Additionally, they look for differences between tissues/germline/embryonic samples in non-methylated regions using BioCAP to look for evidence of NMI in promoters/regulator regions, and further use nanopore sequencing to assess global methylation patterns of germline, embryonic and adult somatic tissues. The authors find the global methylation pattern of eggs, muscle, brain and PBMC's are similar, while those of the sperm, and day1 and day2 embryo are similar, leading them to infer that the embryonic methylation marks are inherited from the paternal parent. This pattern is similar to that found in zebrafish but different from that in mammals. They further find that the chromosomal regions eliminated from somatic cells are hypermethylated in sperm, which suggests that the hypermethylation of these regions may play a role in PGR itself. A less clear part of the paper is their inference that there is a shift in the regions that exhibit partial methylation (PMD's) in sperm and embryo as discussed below.

The authors are clearly experts in DNA methylation analysis, and overall the results are interesting, and the quality of writing fine. I have some comments regarding the interpretation of some of the data, and think that several of the figures would benefit from being modified and having better legends.

General comment - one of the novel things in the paper that the authors describe that they independently estimated the chromosomal regions eliminated during PGR in sea lamprey - and compared these regions to those estimated by Smith et al., 2018. It would be good if the authors could make available as a track the regions they identified as being in the eliminated regions if possible for viewing on UCSC, or give more information about this in the supplementary data. The regions identified as having DMR are nicely laid out in the supp data - not sure why the eliminated regions were not.

I wasn't sure why the authors mapped their data to petMar3. The newer VGP genome is almost a chromosomal level assembly so it would have been nice to see this data related to the newer assembly. I understand that the VGP assembly was under embargo until recently, so perhaps this is the reason. In any case, it would be appropriate to publish the coordinates of the genome the authors found to be eliminated during PGR relative to petMar3, as these will be able to be lifted over.

Did the authors find evidence that the egg also harbours the eliminated regions (as in sperm)? From

their data it appears that it does not, or at least at fertilization shows no evidence of hypermethylation of the region. Some more comment on this would be interesting.

Was the same extraction protocol given on page 11 used for egg and embryo's as well? These are notoriously harder to extract from, although the protocol looks good.

A few other issues were overall unclear to me; one was regarding the PMD's. The authors propose that sperm have more variable PMD, and in particular that the eliminated regions are hypermethylated. The latter inference is clearly supported by the data as I interpreted it, but I don't clearly see the evidence the authors statement that ~ 30% of the lamprey genome transitions to a PMD state post-fertilization. In particular, I found the Figure 5 confusing. All of the figures clearly show that the egg, brain, muscle and PBMC's have similar mCG/CG patterns, while the sperm, day1, day2 embryos are similar.. what I don't infer from this figure is the 30% change in PMD in early embryonic development because the sperm, day1 and day2 appear very similar - supporting their inference that the post-embryonic zygote adopts the paternal methylation pattern. Can the authors please explain this more clearly. Similar in Figure 1 D, they show the loss (4%) and gain (25%) (is the combination of this where the 30% comes from?), off PMD.. but what I see in that there is ~ 25% more regions that are methylated in sperm, day1, day2.. I don't see a gain of PMD in early embryonic development - please clarify.

One of the main ways that the manuscript could be improved is to improve some of the figures - in particular, the figures such as that in Figure 3C, 4C and 4D. Simple things that would make a big difference in interpretation would be to provide better label for the "Y" axis - not just n=849 (this is not defined in the figure legend either).. to something like n=849 eliminated scaffolds. Other suggestions include things such as the rectangular figures are deceiving because it looks like there is "strip" down the middle but of course this is because the sequences being compared are not the same size as the region of interest. In some cases seeing the "space" on the left and right is informative (e.g. left panel 3C), but other times it is deceiving (eg. 4C, purple graphs).

Similarly, many of the figure legends need expansion - it is not easy to interpret several figures without referring to the text and some information is missing all together.. e.g. the authors have r1 and 2 in several figures, these should be written out as replicate 1 and replicate 2 at least in the first figure legend. Another important example is Figure 2E and 2F.. The importance of the PDR, I assume is to reveal how consistent the methylation was between the two replicates or across tissues? ... It is not clear to me what is meant by the bracket above Figure 2E (PMD, egg, brain, muscle) and the individual boxes of egg, sperm and day1)... What does the bracket refer to ? PDR there.. in egg, sperm and day1? Here neither the figure legend nor the text appear to describe what is captured in this figure.

Figure S3E - the GC content of the core NMI's has a tighter range, but higher GC content than in sperm..is this evidence that they are in likely promoter regions?

Figures 3B, 3H etc... would be helpful to tell the reader that the rows refer to the selected number of partitions/clusters (k) indicated by the analyses.

I found supplementary Figure S4 particularly interesting, and only S4E is cited! Several questions here: S4A - the top panel shows the mCG/CG of 14209 DMR's and the bottom panel the same thing but in 10,901 DMR's.. Why are the distributions so different?

Do you have two sets of everything for each biorep in this Figure S4C, 4D etc? please indicate in the legend if so..If so, there should be four comparisons (2eggx2sperm) not two.. how did you choose which ones to compare? The egg-sperm stats between the two comparisons are quite different.. reasons? In S4F - their methylation profiles look *very* different - please comment on what the authors think is going on here (perhaps it just needs a better figure legend).

Can the authors comment on why they see such a difference between the nanopore and WGBS.. Presumably the nanopore is more accurate because it is more genome-wide and less adulterated DNA - or is there another interpretation?

A recent paper found evidence that the eliminated regions are highly expressed during male gonad development - (PMID: 35538209) - seems relevant to cite this as well.

Why did the authors choose 848 overlapping regions? -- do you mean overlapping with Smith et al?

In figure S5B - Were no genic regions identified in the 849 regions? All of the boxes refer to repetitive or low-complexity region yet many genes have been identified there.

Figure S5C - I find the rectangular box figures misleading for these data.. since not all reads are the same length - can't think of a better plot image.. but a better legend would make this less confusing.. noting that all sampled regions are of different sizes.

Reviewer #2 (Remarks to the Author):

Bogdanovic et al. generated a set of DNA methylome datasets in lamprey, a representative species for jawless vertebrates that bridges invertebrates and jawed vertebrates. The authors discovered that Partially Methylated Domains (PMDs) occupy a large fraction of the lamprey genome, and PMDs are highly dynamic during early lamprey embryogenesis. Lamprey also presented another example of maternal-to-paternal DNA methylation transition after fertilization besides zebrafish and medaka. Finally, the authors analyzed methylation states in genomic regions lost during development (programmed genome re-arrangement) and found they bear high methylation in sperm.

Overall, this is a highly valuable resource both for the developmental biology and epigenetics fields to understand DNA methylation evolution and its reprogramming in early embryogenesis. Nevertheless, certain analyses are relatively shallow which hinder a deeper understanding of the underlying biology. Some parts of the manuscript are difficult to understand. These need to be improved before this paper is considered for publication and I have a few suggestions below to help with this.

1. "Moreover, we found that NMIs and hypomethylated DMTs in egg, brain, muscle and PBMCs overlap genic regions to a greater extent than those in sperm, day1, day2, indicative of ZGA-associated epigenome remodeling." This is inaccurate as no samples were collected immediately after ZGA. Brain/muscle/PBMCs are somatic cells which are too far from ZGA. I encourage the authors to add a post-ZGA methylome (such as day 3?) which would greatly raise the value of this study.
2. Most analyses in the paper lack snapshots and gene examples. Adding them would greatly help the readers to understand these figures and results.
3. The paper lacks a general analysis of the relationship between DNA methylome and gene expression. Does DNA methylation level show correlation with gene expression level in lamprey at different stages? Understandably, such analysis is less informative for sperm/ocyte/pre-ZGA embryos.
4. The analysis divided the genome into PMD and non-PMD. However, PMD is usually used to refer to regions with relatively low methylation in the genome in jawed vertebrates such as zebrafish and mammals. Here, it was used to refer to regions with relatively high methylation, which could cause some confusion. Importantly, it appears there are also fully methylated domains (FMDs) in the genome (for example, Fig. 2D, E). I suggest that the authors could consider dividing the genomes into

FMDs, PMDs, and unmethylated domains, which would be more accurate and less confusing.

5. My impression is that after fertilization, most of the transitions are from non-methylated domain to PMDs, but no analysis is presented. A related question is that why global DNA methylation level slightly decrease after fertilization, given that most transitions seem to be non-methylated domain to PMD?

6. Fig. 2E, the sperm/day1/2-specific PMD groups were defined as the "PMD gain" group. However, DNA methylation levels seem comparable between sperm and oocyte, which is different from what the example shows in Fig. 2D. This needs to be clarified. The accompanying statement "whereas the second group, characterized by developmental PMD gain, displayed an increase in intermediate mCG levels, with no apparent changes in low or high mCG fractions" is also unclear what it means.

8. The maternal-to-paternal transition appears to be highly conserved and is one of the most fascinating and mysterious phenomenon in methylome reprogramming in early embryos. Does this happen mainly at distal enhancer sites or TSS sites in lamprey?

9. The related discussion is well written and thoughtful. On the other hand, it would be beneficial to point out in Discussion or Introduction that in zebrafish, the maternal-to-paternal transition does not depend on the paternal genome (Potok et al.). Otherwise the paper could bring the readers the impression that the maternal methylome is "copying" the paternal methylome. It could be helpful to mention another alternative possibility that the sperm methylome already adopts the embryo-like methylome prior to fertilization, while the maternal genome only does so after fertilization (see Wu et al., Science Advances, 2021).

10. In zebrafish, enhancers in sperm and early embryo are fully methylated, which partially contribute to the maternal-to-paternal transition (Wu et al., Science Advances, 2021). Does this also occur in lamprey? I understand that enhancer information may not be available in lamprey. Alternatively, this could be discussed in Discussion. Regardless, it is probably true that none of these models can fully explain this mysterious transition.

11. Page 2. "... Whether 5mC may contribute to the targeting and eliminating of germline-specific genes during PGR". Page 7. "Our results suggest that mCG might play a role in the selection or perhaps even germline protection of eliminated DNA". This is a bit overstretching unless the authors can present related evidence either from lamprey or other species that DNA methylation is involved in PGR.

REVIEWER #1

Overall, this is an interesting paper by Angeloni et al on changes in DNA methylation during early lamprey embryogenesis. The Bogdanovic lab has pioneered new approaches to studying changes in methylation patterns in vertebrates during early embryogenesis. Broadly, there is evidence of major change in global early developmental methylation from invertebrates to mammals, with invertebrates showing very low levels of methylation, while mammalian genomes are hypermethylated. Additionally, there are major differences across vertebrates in the re-programming of methylation marks in early embryogenesis with mammals exhibiting nearly complete erasure of the gametic methylation patterns while in zebrafish, the embryo inherits the methylation pattern found in the sperm. Lying at the base of the vertebrate evolutionary tree, sea lamprey are thus an interesting taxa in which to study global methylation across tissues and developmental stages, as well as the inheritance (or lack thereof) of methylation patterns following fertilization. Furthermore, since lamprey undergo a process termed programmed genome rearrangement (PGR), it is of interest to assess if the regions of the lamprey genome that are eliminated during PGR have differential methylation from the non-eliminated regions.

Angeloni et al., use whole genome bisulfite sequencing (WGBS) to assess global differences in methylation in somatic tissues - muscle, brain and PBMC's, as well as gametes (eggs and sperm) and embryos 1 and 2 dpf, since PGR is complete between days 2.5-3. Additionally, they look for differences between tissues/germline/embryonic samples in non-methylated regions using BioCAP to look for evidence of NMI in promoters/regulator regions, and further use nanopore sequencing to assess global methylation patterns of germline, embryonic and adult somatic tissues. The authors find the global methylation pattern of eggs, muscle, brain and PBMC's are similar, while those of the sperm, and day1 and day2 embryo are similar, leading them to infer that the embryonic methylation marks are inherited from the paternal parent. This pattern is similar to that found in zebrafish but different from that in mammals. They further find that the chromosomal regions eliminated from somatic cells are hypermethylated in sperm, which suggests that the hypermethylation of these regions may play a role in PGR itself. A less clear part of the paper is their inference that there is a shift in the regions that exhibit partial methylation (PMD's) in sperm and embryo as discussed below.

The authors are clearly experts in DNA methylation analysis, and overall the results are interesting, and the quality of writing fine. I have some comments regarding the interpretation of some of the data, and think that several of the figures would benefit from being modified and having better legends.

RESPONSE 1: We thank the Reviewer for the constructive comments and support of the manuscript.

General comment - one of the novel things in the paper that the authors describe that they independently estimated the chromosomal regions eliminated during PGR in sea lamprey - and compared these regions to those estimated by Smith et al., 2018. It would be good if the authors could make available as a track the regions they identified as being in the eliminated regions if possible for viewing on UCSC, or give more information about this in the supplementary data. The regions identified as having DMR are nicely laid out in the supp data - not sure why the eliminated regions were not.

RESPONSE 2: The 849 sequences that we classified as being eliminated are now added as Supplementary Table S19.

I wasn't sure why the authors mapped their data to petMar3. The newer VGP genome is almost a chromosomal level assembly so it would have been nice to see this data related to the newer assembly. I understand that the VGP assembly was under embargo until recently, so perhaps this is the reason. IN any case, it would be appropriate to publish the coordinates of the genome the authors found to be eliminated during PGR relative to petMar3, as these will be able to be lifted over.

RESPONSE 3: Indeed, the latest VGP assembly (kPetMar1) was released following the completion of analysis and writing for this manuscript. We have converted the coordinates of eliminated sequences to the kPetMar1 genome and have included these coordinates in the manuscript as Supplementary Table 20, for anyone who may wish to use them. We have also now cited the paper describing the new reference genome (Timoshevskaya et al; Cell Reports 2023).

Did the authors find evidence that the egg also harbours the eliminated regions (as in sperm)? From their data it appears that it does not, or at least at fertilization shows no evidence of hypermethylation of the region. Some more comment on this would be interesting.

RESPONSE 4: We did not see any clear evidence of the eliminated sequences we identified in sperm being present in the oocyte in our datasets. We have now included a heatmap showing read coverage from our egg whole-genome bisulfite sequencing datasets (in two biological replicates) at eliminated sequences in Supplementary Figure S5D, and the read coverage at these regions is like that of somatic tissues. While this is an interesting and important point to consider, we do not feel that we can make conclusive statements about PGR in egg based on our data alone. We have, however added the following statement to our discussion:

“Currently, genomic and epigenetic studies of PGR in lamprey have mostly been limited to analyses of the male germline. However, in some species, such as the zebra finch, the germline-restricted chromosome (GRC), which is found in both the male and the female germline, is eliminated from mature sperm^{82,83}. In our study, we found that WGBS read coverage at eliminated sequences in the egg was comparable to that of somatic tissues, suggestive of some degree of DNA elimination. Previously, RNA-seq analyses of male and female lamprey gonad development have shown that eliminated genes are 36X more likely to be expressed in testes than in ovaries and that strikingly, ~70% of those genes were not expressed in differentiated ovarian samples⁴³. This indeed suggests that the complement of eliminated genes might differ between the male and female germline and that programmed DNA loss from mature oocytes is indeed a possibility. Nevertheless, given the high content of repetitive DNA at those regions and the fact that our observations are based on WGBS data alone, at this point no conclusive statement can be made regarding the structure and content of PGR sequences in the female germline. Further studies, preferably involving long-read DNA sequencing technologies, will be required to resolve this issue.”

Was the same extraction protocol given on page 11 used for egg and embryo's as well? These are notoriously harder to extract from, although the protocol looks good.

RESPONSE 5: Indeed, the same DNA extraction protocol was used for all samples. The only variation in terms of procedure with respect to egg samples was that the eggs were extensively rinsed under distilled water to remove debris, before subjecting them to DNA isolation. We have updated our DNA extraction protocol in the methods section to clarify this: "All lamprey samples were lysed in buffer containing 20 mM Tris, pH 8.0, 100 mM NaCl, 15 mM EDTA, 1% SDS and 0.5 mg/ml proteinase K for 3 h at 55 °C. Additionally, before DNA extraction, the egg samples were rinsed extensively under distilled water to eliminate debris."

A few other issues were overall unclear to me; one was regarding the PMD's. The authors propose that sperm have more variable PMD, and in particular that the eliminated regions are hypermethylated. The latter inference is clearly supported by the data as I interpreted it, but I don't clearly see the evidence the authors statement that ~ 30% of the lamprey genome transitions to a PMD state post-fertilization. In particular, I found the Figure 5 confusing. All of the figures clearly show that the egg, brain, muscle and PBMC's have similar mCG/CG patterns, while the sperm, day1,day2 embryos are similar.. what I don't infer from this figure is the 30% change in PMD in early embryonic development because the sperm, day1 and day2 appear very similar - supporting their inference that the post-embryonic zygote adopts the paternal methylation pattern. Can the authors please explain this more clearly. Similar in Figure 1 D, they show the loss (4%) and gain (25%) (is the combination of this where the 30% comes from?), off PMD.. but what I see in that there is ~ 25% more regions that are methylated in sperm, day1, day2.. I don't see a gain of PMD in early embryonic development - please clarify.

RESPONSE 6: We thank the Reviewer for pointing out these issues. First, we would like to clarify what we mean when referring to "PMD gain" and "PMD loss" in the embryo, and we appreciate that this point could have been explained better. In line with the rest of the results presented in this manuscript, PMD dynamics follow the maternal-to-paternal reprogramming pattern, thus PMD gain indicates that the maternally contributed genome "gains" partial DNA methylation in the early embryo, undergoing reprogramming to match the paternal genome, before ZGA. In regions undergoing PMD loss, the PMDs on the maternally derived genome are reprogrammed to a non-PMD state in the early embryo, before ZGA. Together, these reprogrammed PMDs make up 29% of the genome. To clarify this issue, we have now included a schematic in Figure 2E, describing four major PMD reprogramming events and the genomic percentage affected by them. Moreover, to avoid confusion we now refer to these reprogramming events as "Maternal PMD gain" and "Maternal PMD loss", as it is indeed only the maternal genome contribution that gains or loses the PMD configuration (Figure 2E). We have also amended Supplementary Figure S2 and have updated Figure 5 to clarify our conclusions. Moreover, to obtain better insight into developmental PMD dynamics, we partitioned the non-PMD genome fraction into: i) unmethylated regions (UMRs), ii) lowly methylated regions (LMRs), and iii) hypermethylated regions (hyperMR - regions that generally contain increased DNA methylation levels compared to the genomic mean and that are not identified as PMDs by the MethylSeekR algorithm) (Supplementary Figure S2). We then plotted for each of these scenarios (Figure 2E) the developmental (maternal to paternal) change in DNA methylation levels (mCG / CG) and read discordancy (PDR - percentage of "partially methylated" reads), now clearly demonstrating for all major reprogramming

scenarios (UMR -> PMD, LMR -> PMD, hyperMR -> PMD, and PMD -> UMR), that significant difference in mCG patterning (both at the average mCG levels, and individual reads – i.e DNA molecules), exists between 1) egg, and 2) sperm, day 1, and day 2 samples (Figure 2F).

One of the main ways that the manuscript could be improved is to improve some of the figures - in particular, the figures such as that in Figure 3C, 4C and 4D. Simple things that would make a big difference in interpretation would be to provide better label for the "Y" axis - not just n=849 (this is not defined in the figure legend either).. to something like n=849 eliminated scaffolds.

RESPONSE 7: We have updated the Y-axis labels for heatmaps for Figure 3B-C, Figure 4C-D, Supplementary Figure S4F and Supplementary Figure S5C-E (now Supplementary Figure S5D-F) to clarify which regions are being depicted in the heatmaps.

Other suggestions include things such as the rectangular figures are deceiving because it looks like there is "strip" down the middle but of course this is because the sequences being compared are not the same size as the region of interest. In some cases seeing the "space" on the left and right is informative (e.g. left panel 3C), but other times it is deceiving (eg. 4C, purple graphs).

RESPONSE 8: Figure 4C-D and Supplementary Figure S5C-E (now Supplementary Figure S5D-F) initially showed eliminated sequences scaled to the same size. We have replaced these heatmaps with regions unscaled, to better represent the size distribution of eliminated sequences. We have also updated our methods (under "Identification of sequences eliminated during PGR" and "mCG at eliminated sequences") to describe how these heatmaps were generated.

Similarly, many of the figure legends need expansion - it is not easy to interpret several figures without referring to the text and some information is missing all together.. e.g. the authors have r1 and 2 in several figures, these should be written out as replicate 1 and replicate 2 at least in the first figure legend.

RESPONSE 9: We have now clarified in our figure legends whether the panel is representing two individual replicates or both replicates merged. In cases where only one replicate is shown in the main figure, we have specified this in the figure legend and have included both replicates in supplementary figures and refer to both main and supplementary figures when discussing our findings in our text. We have also clarified that r1 and r2 mean replicate 1 and replicate 2 respectively in our figure legends.

Another important example is Figure 2E and 2F.. The importance of the PDR, I assume is to reveal how consistent the methylation was between the two replicates or across tissues?

RESPONSE 10: We thank the Reviewer for pointing this out. Indeed, we agree that this part could have been explained better. The purpose of the PDR analysis was to compare read discordancy at reprogrammed PMDs between tissues, so that we could better decipher 5mC dynamics at these sites of highly heterogeneous 5mC. We have now clarified this in our

manuscript: “As the regions undergoing reprogramming are partially methylated in either egg or sperm, we next took advantage of our single-molecule resolution data to study how this 5mC heterogeneity is generated at the individual read level, with the aim of resolving how partial 5mC states are established and lost during early pre-ZGA embryogenesis. To achieve this, we calculated the proportion of discordant reads (PDR – proportion of reads containing both methylated and unmethylated CpG sites) ^{49–51} within PMDs and plotted these values against PMD mCG levels (**Figure 2F**).” Thus plotting PDR against mCG levels in reprogrammed PMDs, allows for visualising DNA methylation changes in bulk and at the single read (i.e single DNA molecule) level. For example, see **Review Figure 1** (below). Figure 2F has been amended to show all four major reprogramming scenarios and how DNA methylation changes between 1) egg, and 2) sperm, day1, and day2.

Review Figure 1. Proportion of discordant reads (PDR). PDR is defined as proportion of discordant (partially methylated) reads. Plotting PDR against DNA methylation (mCG/CG) levels of genomic features allows us to obtain insight into the methylation state of single DNA molecules (reads). For example, an average mCG/CG value of 0.5 might correspond to two scenarios: 1) All reads covering the feature are 50% methylated (discordant), or the reads covering the feature consist of 50% fully methylated (concordant) reads and 50% unmethylated (concordant) reads. In scenario 1) The PDR value would be high (i.e close to 1) as all the reads would be discordant (partially methylated at 50%) whereas in scenario 2) the discordancy would be low (i.e close to 0), as all the reads would be concordant (either fully methylated or fully unmethylated). In the example above, PMDs in egg, on average (see red line), have intermediate mCG levels (~0.3) and a high PDR (~0.7), suggesting that the partially methylated (PMD) state is characterised by many partially methylated (discordant reads), rather than a mixture of fully methylated and fully unmethylated (concordant) reads covering these genomic features. In sperm, the regions corresponding to the shown subset of egg PMDs, exist in an unmethylated (UMR) state, characterised by low mCG levels (i.e < 0.1) and the read discordancy is low (~0.2) suggestive of a large number of fully unmethylated (concordant) reads covering these features.

It is not clear to me what is meant by the bracket above Figure 2E (PMD, egg, brain, muscle) and the individual boxes of egg, sperm and day1)... What does the bracket refer to ? PDR there.. in egg, sperm and day1? Here neither the figure legend nor the text appear to describe what is captured in this figure.

RESPONSE 11: Initially, the brackets in Figure 2E (now Figure 2F) were intended to represent PMDs that were present in egg and differentiated somatic tissues, but not present in sperm or early embryonic tissues (i.e. PMDs that are reprogrammed). We understand that these brackets are confusing and have replaced these titles with “Maternal PMD gain” and “Maternal PMD loss”. We have now included a schematic in Figure 2E that explains PMD reprogramming, and what is meant by “PMD gain” and “PMD loss”. Figure 2F (previously Figure 2E) shows PDR and mCG levels at reprogrammed PMDs, with each box representing

PDR and mCG values for individual tissues. For more information on PDR, please see Review Figure 1 (above).

Figure S3E - the GC content of the core NMI's has a tighter range, but higher GC content than in sperm..is this evidence that they are in likely promoter regions?

RESPONSE 12: Although we see BioCAP signal at transcription start sites across all tissues (Supplementary Figure S3E), we do not see greater overlap between core NMIs and promoters when compared to the overlap between tissue-specific NMIs and promoters (Figure 3F).

Figures 3B, 3H etc... would be helpful to tell the reader that the rows refer to the selected number of partitions/clusters (k) indicated by the analyses.

RESPONSE 13: We have included an additional y-axis title for Figure 3B and H (now J) and Supplementary Figure S3B that shows the number of k clusters for these heatmaps. We have also updated the figure legends for these panels with the following: "Y-axis boxes refer to individual clusters."

I found supplementary Figure S4 particularly interesting, and only S4E is cited! Several questions here: S4A - the top panel shows the mCG/CG of 14209 DMR's and the bottom panel the same thing but in 10,901 DMR's. Why are the distributions so different?

RESPONSE 14: In Figure S4A, the first set of DMRs ($n = 14,209$) refers to DMRs hypermethylated in egg and hypomethylated in sperm (as well as in day1 and day2 samples). The second set of DMRs ($n = 10,901$) refers to DMRs hypermethylated in sperm and hypomethylated in egg (as well as in brain, muscle, and PBMC samples). The color code (blue = hypomethylation, red = hypermethylation) is now explained in the supplementary figure legend. As for the difference in the mCG distribution, we believe that these two sets of elements might exert different developmental functions. For example, given that lamprey displays comparable DNA methylation reprogramming dynamics to zebrafish and medaka, it is possible that the DMRs hypermethylated in egg and hypomethylated in sperm, day1 and day2 samples correspond to hypomethylated regulatory regions associated with the placeholder nucleosome structure in zebrafish (marked by H2A.Z, H3K4me1 and hypomethylation). Placeholder nucleosomes are usually associated with developmental and housekeeping genes required for early embryogenesis and are thus unmethylated in sperm, and early embryos (i.e day1 and day2 in lamprey). On the other hand, the DMRs hypermethylated in sperm and hypomethylated in egg may correspond to enhancers required for organogenesis, which are important for later stages of embryonic development and tissue differentiation. Importantly, these two sets of DMRs display different genomic localisation, with sperm hypermC-DMRs enriched in distal regions (in line with their enhancer function), and somewhat different CpG density distributions, which might have an impact on their methylation state. We have now elaborated on this in the main text (see below) both in the results section and in the discussion (see below).

RESULTS:

"Notably these two major DMR groups were characterized by distinct features, including differences in mCG levels, genomic localization, CpG density distribution, PMD content, and

BioCAP signal (Supplementary Figure S4A-F), indicative of different biological functions (see discussion)”

DISCUSSION:

“It remains elusive exactly how mCG remodeling is achieved in lamprey. Based on mCG reprogramming dynamics identified in our study, it is not unlikely that lamprey utilize placeholder nucleosomes enriched in H2A.Z and H3K4me1 in a similar manner to zebrafish²⁸. These placeholder nucleosomes are specific to the paternal germline in zebrafish and establish pre-ZGA chromatin in the early embryo. In zebrafish, placeholder nucleosomes are found at hypomethylated regulatory regions associated with developmental and housekeeping genes, where they deter DNMT activity while maintaining a transcriptionally quiescent state during cleavage stages²⁸. Thus, it is plausible that NMIs and hypomethylated DMRs in sperm and embryonic tissues, represent sequences associated with a similar chromatin configuration in lamprey, which consequently display reduced mCG levels compared to tissues where such nucleosome positioning may not be present. In terms of DMRs that we find hypermethylated in lamprey sperm and embryonic tissues, a recent zebrafish study identified that DNA hypermethylation of CpG-rich enhancers in sperm and pre-ZGA embryos safeguards embryonic programs by preventing premature activation of transcriptional programs associated with adult tissues⁶⁷. Depletion of *dnmt1* resulted in severe developmental defects and embryonic lethality that was linked to ectopic activation of these enhancers, emphasizing that inheritance and maintenance of paternal-like 5mC states plays a critical role in regulating developmental programs. It is important to note that in zebrafish, establishment of the embryonic epigenome is not dependent on the paternal genome; even in parthenogenetic embryos (maternal haploids that lack sperm DNA), a sperm-like chromatin configuration is still observed^{27,28,67}. This suggests that in zebrafish, and probably also lamprey, the embryonic methylome is already established in transcriptionally quiescent sperm, potentially to ease the establishment of totipotency by only reprogramming one parental allele.”

Do you have two sets of everything for each biorep in this Figure S4C, 4D etc? please indicate in the legend if so..If so, there should be four comparison (2eggx2sperm) not two.. how did you chose which ones to compare? The egg-sperm stats between the two comparisons are quite different.. reasons?

RESPONSE 15: We have now clarified in our figure legends whether the panel is representing two individual replicates or both replicates merged. In cases where only one replicate is shown in the main figure, we have specified this in the figure legend and have included both replicates in supplementary figures and refer to both main and supplementary figures when discussing our findings in our text. We have also clarified that r1 and r2 means replicate 1 and replicate 2 respectively in our figure legends. For some analyses (namely PMD identification and DMR calling), it was necessary to merge WGBS replicates. We have now updated our methods to state whether a particular analysis used merged WGBS replicates. The egg-sperm stats between the two comparisons are different as they correspond to different DMR types (sperm-hyper, egg-hypo methylated; sperm-hypo, egg-hypermethylated). We have now clarified this in the figure legend.

In S4F - they methylation profiles look *very* different - please comment on what the authors think is going on here (perhaps it just needs a better figure legend).

Can the authors comment on why they see such a difference between the nanopore and WGBS.. Presumably the nanopore is more accurate because it is more genome-wide and less adulterated DNA - or is there another interpretation?

RESPONSE 16: We assume the Reviewer is referring to Supplementary Figure S5F (now Supplementary Figure S5G). We find that 5mC levels in Nanopore data are hypermethylated (median >75% mCG/CG) compared to the global mean, but appear more heterogeneous compared to WGBS. This is most likely because there is greater coverage of CpG sites in eliminated sequences in Nanopore compared to short-read WGBS data. We have now included the following comment under “DNA sequences eliminated during PGR are hypermethylated in sperm” in our Results section:

“Long-read sequencing is particularly useful for this analysis as eliminated sequences contain repetitive elements that may not be as well covered by short-read datasets (Supplementary Figure S5C).”

Nevertheless, eliminated sequences are hypermethylated in both datasets, so we do not feel that these differences in mCG distribution impact our conclusion. We have also now included a panel (S5H) that shows correlation between 5mC levels at single CpG sites in sperm Nanopore 5mC biological replicates and sperm WGBS 5mC biological replicates. In this panel, CpG sites used for plotting are covered by all four libraries, at least 5X (n = 50693 CpG sites). We identify a strong positive relationship ($R > 0.75$) between libraries in pairwise comparisons, and show that the majority of CpG sites are hypermethylated in all four datasets.

A recent paper found evidence that the eliminated regions are highly expressed during male gonad development - (PMID: 35538209) - seems relevant to cite this as well.

RESPONSE 17: This is now added and discussed. See RESPONSE 4

Why did the authors choose 848 overlapping regions? -- do you mean overlapping with Smith et al?

RESPONSE 18: Yes, the 848 sequences are those that are common to both our dataset of sequences identified with CNVkit and to those previously published by Smith et al. We have now clarified this in our manuscript:

“We then selected 849 regions (~9.3Mb) overlapping both eliminated sequences identified by CNVkit and those previously published (J. J. Smith et al. 2018) as a stringent dataset for further genome-scale analyses (Figure 4A-B, Supplementary Figure S5A, Supplementary Table S19-20).”

In figure S5B - Were no genic regions identified in the 849 regions? All of the boxes refer to repetitive or low-complexity region yet many genes have been identified there.

RESPONSE 19: We have now added a panel in Supplementary Figure S5 (panel B) which shows the overlap between genes within our CNVkit eliminated regions, those within previously published eliminated sequences (Smith et al 2018) and genes common to both datasets. We have also included the genes common to both datasets as a supplementary table (Supplementary Table S21). The function of these genes has been studied elsewhere, so we did not feel that additional analysis was beneficial for our study. We have now included the following statement in our manuscript:

“These overlapping eliminated sequences contain 277 genes, the function of which have been described elsewhere ^{42–44} (**Supplementary Figure S5B, Supplementary Table S21**).”

Figure S5C - I find the rectangular box figures misleading for these data.. since not all reads are the same length - can't think of a better plot image.. but a better legend would make this less confusing.. noting that all sampled regions are of different sizes.

RESPONSE 20: We thank the Reviewer for pointing this out. We have now replaced Supplementary Figure S5C (now Supplementary Figure S5D) with a heatmap depicting regions unscaled to better represent the size distribution of eliminated sequences. Also, see RESPONSE 8.

REVIEWER #2

Bogdanovic et al. generated a set of DNA methylome datasets in lamprey, a representative species for jawless vertebrates that bridges invertebrates and jawed vertebrates. The authors discovered that Partially Methylated Domains (PMDs) occupy a large fraction of the lamprey genome, and PMDs are highly dynamic during early lamprey embryogenesis. Lamprey also presented another example of maternal-to-paternal DNA methylation transition after fertilization besides zebrafish and medaka. Finally, the authors analyzed methylation states in genomic regions lost during development (programmed genome re-arrangement) and found they bear high methylation in sperm.

Overall, this is a highly valuable resource both for the developmental biology and epigenetics fields to understand DNA methylation evolution and its reprogramming in early embryogenesis. Nevertheless, certain analyses are relatively shallow which hinder a deeper understanding of the underlying biology. Some parts of the manuscript are difficult to understand. These need to be improved before this paper is considered for publication and I have a few suggestions below to help with this.

RESPONSE 1: We thank the Reviewer for the constructive comments and support of the manuscript.

1. “Moreover, we found that NMIs and hypomethylated DMTs in egg, brain, muscle and PBMCs overlap genic regions to a greater extent than those in sperm, day1, day2, indicative of ZGA-associated epigenome remodeling.” This is inaccurate as no samples were collected immediately after ZGA. Brain/muscle/PBMCs are somatic cells which are too far from ZGA. I encourage the authors to add a post-ZGA methylome (such as day 3?) which would greatly raise the value of this study.

RESPONSE 2: We thank the Reviewer for pointing this out and we can certainly understand how this might sound confusing. Following the Reviewer's suggestion, we have decided to remove this sentence to reflect our conclusions more accurately. We agree that an additional embryonic post-ZGA methylome (i.e day 3 – gastrula) could possibly raise the value of this study, as it could allow for a direct comparison of mCG and transcription. Unfortunately, however, we do not currently have access to these samples. Also, for comparisons such as these ones, we would prefer to have all the embryos collected from the same single batch, or from the same mixture of batches to minimise the effect of genetic and consequently epigenetic heterogeneity. Given the exciting results from the current study, in the near future, we are hoping to conduct follow-up experiments, including simultaneous transcriptome and epigenome profiling at single-cell resolution of later embryonic stages. We are currently looking into the possibility of obtaining permits and conducting lamprey research in Spain via collaborators. Also, to study the effects of 5mCG on transcriptional repression in the lamprey we took advantage of a previously published adult brain transcriptome dataset and compared it to our brain 5mC data (see RESPONSE 4 for more details).

2. Most analyses in the paper lack snapshots and gene examples. Adding them would greatly help the readers to understand these figures and results.

RESPONSE 3: We thank the Reviewer for this suggestion. We have updated our genome browser screenshots for the PMD transitions, now showing examples of all major transitions (PMD-UMR, UMR-PMD, hyperMR-PMD, LMR-PMD) in Figure S2D. Moreover, we have added gene examples for core NMIs, adult specific NMIs, and embryonic NMIs to Figure 3D, as well as examples of major DMR groups (hypermC sperm-egg, and hypermC egg-sperm - Figure 3H). More examples of the lamprey DNA methylome landscape have been added to Figure S1C.

3. The paper lacks a general analysis of the relationship between DNA methylome and gene expression. Does DNA methylation level show correlation with gene expression level in lamprey at different stages? Understandably, such analysis is less informative for sperm/oocyte/pre-ZGA embryos.

RESPONSE 4: We have now included an analysis on the relationship between promoter DNA methylation and gene expression in our manuscript (Supplementary Figure S4G-H). Overall, we do found a weak, but negative correlation between promoter DNA methylation levels and gene expression. We have included a comment on this in our manuscript: "Finally, we wanted to assess whether DMRs and differentially enriched NMIs overlapping regulatory elements were linked to expression changes at corresponding genes. As 5mC-mediated CGI silencing is most commonly characterized by gain of somatic mCG at gene promoters¹⁰, we focused our analysis on promoters of protein-coding genes overlapping an NMI ($n = 5188$ genes), and assessed the correlation between NMI 5mC levels and transcription for an adult somatic tissue (brain) (**Supplementary Figure S4G**)⁶⁰. Overall, in line with canonical 5mC function, we identified a weak negative correlation between 5mC and transcription ($r = -0.23$). To study the dynamics of 5mC-mediated gene silencing during development, we next identified promoters of protein-coding genes overlapping both i) regions hypermethylated in egg/brain/muscle/PBMC compared to sperm/day1/day2, and ii) NMIs enriched in sperm/day1/day2 compared to brain/muscle/PBMC ($n = 13$ genes) (**Supplementary Figure S4H**). We then compared the expression profiles of these genes in

diverse embryonic and adult somatic tissues⁶⁰. Again, we observed an overall negative correlation between the presence of mCG and absence of NMI signal in adult tissues, and transcriptional activity, which was most notable in adult brain, kidney, and liver tissues (**Supplementary Figure S4H**). Nevertheless, it is worth pointing out that relationships between promoter mCG and transcriptional silencing are complex^{55,61,62} and that even in organisms with highly methylated genomes, like zebrafish and medaka^{10,26,27,29}, such clear anti-correlation can only be observed on a small number of genes. Altogether, our results indicate discrete mCG remodeling occurring not only within, but also alongside large-scale genomic transitions in PMD state during early development, affecting at least 2% of the genome.”

4. The analysis divided the genome into PMD and non-PMD. However, PMD is usually used to refer to regions with relatively low methylation in the genome in jawed vertebrates such as zebrafish and mammals. Here, it was used to refer to regions with relatively high methylation, which could cause some confusion.

RESPONSE 5: We have added the following statement to our manuscript to clarify this: “It is worth pointing out that while PMDs in heavily methylated (>80% mCG) mammalian genomes typically display reduced 5mC levels compared to the global mean⁴, PMDs in lamprey had heterogenous, but increased 5mC compared to non-PMD regions that generally showed low levels of 5mC (Figure 2A), with the exception of hyperMRs (Supplementary Figure S2B).”

Importantly, it appears there are also fully methylated domains (FMDs) in the genome (for example, Fig. 2D, E). I suggest that the authors could consider dividing the genomes into FMDs, PMDs, and unmethylated domains, which would be more accurate and less confusing.

RESPONSE 6: We thank the Reviewer for this suggestion. For analyses that include a comparison between PMDs and non-PMDs, we have now divided our non-PMD fraction into: i) unmethylated regions (UMRs), ii) lowly methylated regions (LMRs), and iii) hypermethylated regions (hyperMR - regions that generally contain increased DNA methylation levels compared to the genomic mean and that are not called as PMDs by MethylSeekR). We have updated Figure 2, Supplementary Figure S2 and the main text to elaborate on these changes. Indeed with the non-PMD fraction divided in these three different fractions now, the transitions to the PMD state are much cleaner (see Figure 2E-F). The entire section “Maternal-to-paternal embryonic reprogramming of partial DNA methylation” has now been rewritten to accommodate these changes.

5. My impression is that after fertilization, most of the transitions are from non-methylated domain to PMDs, but no analysis is presented.

RESPONSE 7: Indeed, as the Reviewer notes, most transitions are to the PMD state. As per Reviewer’s suggestion, we have now re-analysed these datasets with the non-PMD genome fraction partitioned in UMRs, LMRs, and hyperMRs. We have added a schematic in Figure 2E demonstrating all the major changes and the percentage of the genome affected, whereas detailed analyses are presented in Figure 2F. Briefly, 25% of the (maternal) genome transitions to the PMD state during early embryogenesis. The exact percentages are: UMR-PMD = 16%; hyperMR to PMD = 7%; and LMR-PMD = 2%. Conversely, 4% of the genome loses PMD in the

PMD-UMR transition. The entire section “Maternal-to-paternal embryonic reprogramming of partial DNA methylation” has been rewritten according to these updated results.

A related question is that why global DNA methylation level slightly decrease after fertilization, given that most transitions seem to be non-methylated domain to PMD?

RESPONSE 8: Our new results (see replies to previous two comments), offer a possible explanation for this slight dip. We believe that this could be due to hyperMRs transitioning to the PMD state (7%) and some PMDs being reprogrammed to UMRs (4%). Both transitions are associated with net mCG loss. It is thus possible that the maternal to paternal reprogramming is not perfect i.e that pre-ZGA embryos largely resemble the paternal state in terms of global methylome and PMD structure, however they do not yet reach the global mCG levels observed in sperm (or egg, and other adult somatic tissues), that might increase during organ differentiation and aging.

6. Fig. 2E, the sperm/day1/2-specific PMD groups were defined as the “PMD gain” group. However, DNA methylation levels seem comparable between sperm and oocyte, which is different from what the example shows in Fig. 2D. This needs to be clarified.

The accompanying statement “whereas the second group, characterized by developmental PMD gain, displayed an increase in intermediate mCG levels, with no apparent changes in low or high mCG fractions” is also unclear what it means.

RESPONSE 9: We have now re-analysed these data according to the Reviewers’ suggestions by partitioning the non-PMD fraction into UMRs, LMRs and hyperMRs. The panels to which the Reviewer refers are now replaced with Figure 2E-F that demonstrate four major reprogramming scenarios. The entire section “Maternal-to-paternal embryonic reprogramming of partial DNA methylation” has been rewritten based on these new results.

8. The maternal-to-paternal transition appears to be highly conserved and is one of the most fascinating and mysterious phenomenon in methylome reprogramming in early embryos. Does this happen mainly at distal enhancer sites or TSS sites in lamprey?

RESPONSE 10: We find that DMRs hypomethylated in egg and adult somatic tissues, and NMIs enriched in adult somatic tissues, are more frequently found at genic regions, while hypomethylated DMRs and NMIs in sperm and embryonic tissues are more commonly found in intergenic regions (Figure 3F-G, Supplementary Figure S4C). To better understand what underlies maternal-to-paternal reprogramming, more detailed epigenomic maps of histone marks and chromatin accessibility during developmental stages should be considered. The third paragraph of our discussion explores potential mechanisms of reprogramming in lamprey, with a focus on gene regulatory elements. Also, see RESPONSE 14 to Reviewer 1.

9. The related discussion is well written and thoughtful. On the other hand, it would be beneficial to point out in Discussion or Introduction that in zebrafish, the maternal-to-paternal transition does not depend on the paternal genome (Potok et al.). Otherwise the paper could bring the readers the impression that the maternal methylome is “copying” the paternal

methylome. It could be helpful to mention another alternative possibility that the sperm methylome already adopts the embryo-like methylome prior to fertilization, while the maternal genome only does so after fertilization (see Wu et al., Science Advances, 2021).

RESPONSE 11: We thank the Reviewer for this suggestion. We have added the following to our discussion. “It is important to note that in zebrafish, establishment of the embryonic epigenome is not dependent on the paternal genome; even in parthenogenetic embryos (maternal haploids that lack sperm DNA), a sperm-like chromatin configuration is still observed (Murphy et al. 2018; Potok et al. 2013; Wu et al. 2021). This suggests that in zebrafish, and probably also lamprey, the embryonic methylome is already established in transcriptionally quiescent sperm, potentially to ease the establishment of totipotency by only reprogramming one parental allele.”

10. In zebrafish, enhancers in sperm and early embryo are fully methylated, which partially contribute to the maternal-to-paternal transition (Wu et al., Science Advances, 2021). Does this also occur in lamprey? I understand that enhancer information may not be available in lamprey. Alternatively, this could be discussed in Discussion. Regardless, it is probably true that none of these models can fully explain this mysterious transition.

RESPONSE 12: Given the extent of lamprey reprogramming, and the similarities to zebrafish and medaka, we believe that this might indeed be a possibility. This is supported by distinct sequence features and genomic localisation of egg and sperm DMRs. We have now added the following to the results and discussion sections:

Results: “Notably these two major DMR groups were characterized by distinct features, including differences in mCG levels, genomic localization, CpG density distribution, PMD content, and BioCAP signal (Supplementary Figure S4A-F), indicative of different biological functions (see discussion).”

Discussion: “In terms of DMRs that we find hypermethylated in lamprey sperm and embryonic tissues, a recent zebrafish study identified that DNA hypermethylation of CpG-rich enhancers in sperm and pre-ZGA embryos safeguards embryonic programs by preventing premature activation of transcriptional programs associated with adult tissues (Wu et al. 2021). Depletion of *dnmt1* resulted in severe developmental defects and embryonic lethality that was linked to ectopic activation of these enhancers, emphasizing that inheritance and maintenance of paternal-like 5mC states plays a critical role in regulating developmental programs.”

11. Page 2. “... Whether 5mC may contribute to the targeting and eliminating of germline-specific genes during PGR”. Page 7. “Our results suggest that mCG might play a role in the selection or perhaps even germline protection of eliminated DNA”. This is a bit overstretching unless the authors can present related evidence either from lamprey or other species that DNA methylation is involved in PGR.

RESPONSE 13: Although we have cited and discussed studies where it is suggested that 5mC is implicated in removal of germline sequences (Timoshevskiy et al. PLoS Genetics 2016), we

agree that both these sentences overstate our findings and we have rewritten them in our manuscript:

“...whether sequences eliminated during PGR might be characterized by distinct 5mC patterning.”

“Although further research is necessary to clarify this, it is possible that mCG contributes to the selection or perhaps even germline protection of eliminated DNA, underscoring the significance of mCG dynamics in diverse developmental processes.”

REVIEWER COMMENTS

[Editor: please see attached document for comments from Reviewer 1.]

Overall, Angeloni have addressed my prior concerns. I have one major outstanding concern regarding the revisions, but am otherwise think that the revisions substantially improved the manuscript. Although I am pleased that they lifted over the coordinates of the eliminated regions from the Smith 2018 genome to the newest petmar1 reference genome, it introduced concern about the regions identified as being eliminated. Timoshevskaya et al (2023, in Cell Reports) report the list of eliminated regions identified in petmar1 using their difcover approach, and the regions identified in the current paper vs this Timoshevskaya et al 2023 paper differ substantially.

In their study, @angeloni2023 employed short-read whole-genome sequencing, WGBS, and Nanopore sequencing to examine blood and sperm DNA, aiming to identify genomic sequences that are eliminated during PGR. They utilized CNVkit (@talevich2016) in a whole-genome mode, identifying sequences as eliminated if they had a copy number of 0 in blood relative to sperm. This process requires a bed file containing target genic regions for analysis. However, the authors did not clarify if they used a standard gtf file or a custom set of coordinates for this purpose. Furthermore, the methodology for comparing blood and sperm data remains unspecified. Since CNVkit typically analyzes CNVs using data from a single bam file, the process of comparison between blood and sperm data in their approach remains unclear, as it would not be an automatic feature of the tool.

@timoshevskaya2023 identified germline-specific regions by analyzing ~52X coverage in germline (sperm) reads and ~96X coverage in somatic (blood) reads, using Illumina sequencing technology. They aligned these sequences to the genome assembly with BWA-mem, filtering to retain only primary alignments for further analysis. Then, DifCover was used to assess the extent of germline enrichment across 500bp intervals of low-copy sequences, using modal coverages for sperm and blood. This included masking low coverage areas (below 1/3X in both samples) and high coverage areas (exceeding 3X modal coverage in both). To pinpoint germline-specific genes at higher copy numbers, they modified their DifCover analysis to mask low coverage areas with read depths < 10X and high coverage areas with read depths > 30X modal coverage in both samples. CNVkit and the DifCover both serve the purpose of detecting copy number variations (CNVs) in genomic data, yet they employ distinct methodologies. DifCover calculates average coverage per window and compares it between samples, merging similar windows in contiguous genomes for broader region analysis.

A quick comparison of the results of the current paper vs the results in @timoshevskaya2023, indicates that Angeloni et al. (2023) identified 112 chromosomes/scaffolds covering 5.23 Mb that were eliminated from somatic genome. In contrast, Timoshevskaya et al. (2023) identified 29.119 germline-enriched Mb across 315 chromosomes/scaffolds. The overlap between the two approaches encompassed 4.937 Mb across 101 regions. The assembled eliminated chromosomes from Angeloni et al. (2023) included parts of chromosomes 3, 6, 13, 18, 30, 35, 39, 44, 48, 81. Conversely, Timoshevskaya et al. (2023) identified only chromosomes 51, 81 as well as many scaffolds. Most of the regions identified by CNVkit covered less than 50% of a scaffold/chromosome sequence, whereas DiffCover found that the majority or entirety of a chromosome or scaffold was removed and only three regions had <50% of eliminated sequence: chr 51, NW_022638341.1, NW_022639355.1 (see figure below). Since chromosomal diminution refers to the shortening of chromosomes after programmed genome rearrangement

(PGR), while chromosomal elimination involves removal of entire chromosomes, the results imply different processes of PGR. Since there is other evidence for chromosomal elimination (other papers by the Smith lab) not chromosome diminution, and the results of the Angeloni study are based on WGBS, the results of the Angeloni paper likely identified regions that are differentially methylated not eliminated *per se*. In any case, it would make sense to revise the manuscript accordingly, so as not to introduce confusion into this unresolved issue.

Reviewer #2 (Remarks to the Author):

The authors have well addressed all my comments. I congratulate the authors for this excellent study.

Overall, Angeloni have addressed my prior concerns. I have one major outstanding concern regarding the revisions, but am otherwise think that the revisions substantially improved the manuscript. Although I am pleased that they lifted over the coordinates of the eliminated regions from the Smith 2018 genome to the newest petmar1 reference genome, it introduced concern about the regions identified as being eliminated. Timoshevskaya et al (2023, in Cell Reports) report the list of eliminated regions identified in petmar1 using their difcover approach, and the regions identified in the current paper vs this Timoshevskaya et al 2023 paper differ substantially.

RESPONSE 1: We thank the Reviewer for their positive feedback, and for bringing these issues to our attention. The observed differences are largely due to the limitations associated with lifting over eliminated sequences from the petMar3 genome to the kPetMar1 genome, which we have detailed below. We believe that the best course of action is to remove the supplementary table containing the lifted regions from our manuscript, to prevent any potential misrepresentation of our data.

In their study, @angeloni2023 employed short-read whole-genome sequencing, WGBS, and Nanopore sequencing to examine blood and sperm DNA, aiming to identify genomic sequences that are eliminated during PGR. They utilized CNVkit (@talevich2016) in a whole-genome mode, identifying sequences as eliminated if they had a copy number of 0 in blood relative to sperm. This process requires a bed file containing target genic regions for analysis. However, the authors did not clarify if they used a standard gtf file or a custom set of coordinates for this purpose. Furthermore, the methodology for comparing blood and sperm data remains unspecified. Since CNVkit typically analyzes CNVs using data from a single bam file, the process of comparison between blood and sperm data in their approach remains unclear, as it would not be an automatic feature of the tool.

@timoshevskaya2023 identified germline-specific regions by analyzing ~52X coverage in germline (sperm) reads and ~96X coverage in somatic (blood) reads, using Illumina sequencing technology. They aligned these sequences to the genome assembly with BWA-mem, filtering to retain only primary alignments for further analysis. Then, DifCover was used to assess the extent of germline enrichment across 500bp intervals of low-copy sequences, using modal coverages for sperm and blood. This included masking low coverage areas (below 1/3X in both samples) and high coverage areas (exceeding 3X modal coverage in both). To pinpoint germline-specific genes at higher copy numbers, they modified their DifCover analysis to mask low coverage areas with read depths < 10X and high coverage areas with read depths > 30X modal coverage in both samples. CNVkit and the DifCover both serve the purpose of detecting copy number variations (CNVs) in genomic data, yet they employ distinct methodologies. DifCover calculates average coverage per window and compares it between samples, merging similar windows in contiguous genomes for broader region analysis.

RESPONSE 2: We thank the Reviewer for bringing this to our attention. We would like to acknowledge that the paper by Timoshevskaya *et al* is an important resource and we will certainly be using the latest genome assembly and updated eliminations in future research. We have also now cited this paper in our manuscript, as was suggested by the Reviewer in the first round of reviews. We have now updated our methods section to more clearly describe how we identified eliminated sequences:

Identification of sequences eliminated during PGR

Publicly available blood and sperm whole-genome sequencing reads in FASTQ format were downloaded (J. J. Smith *et al.* 2018). Reads were trimmed using *trimmomatic* and aligned to the *petMar3* reference genome using *bowtie2* (`-p 10 -N 1 --very-sensitive -X 2000 --no-mixed --no-discordant`). Read alignments in BAM format were input into CNVkit (`cnvkit.py -batch -m wgs -f petMar3.fa`) to identify sequences eliminated during PGR, with blood used as input and sperm used as the control sample (`--normal`) (Talevich *et al.* 2016). We defined homozygous deletions as regions with a copy number of 0 in blood. We intersected eliminated sequences identified by CNVkit with a set of previously published eliminated sequences (J. J. Smith *et al.* 2018) as a stringent dataset for further analysis. Sequences were validated using coverage metrics from short-read whole-genome sequencing, WGBS and Nanopore sequencing. Read depth was calculated using *samtools* *depth* function and converted to bigWig format using the *bedGraphToBigWig* script from *kentUtils*. Heatmaps of read coverage at eliminated sequences were generated using *deepTools* *computeMatrix* function (`computeMatrix reference-point --missingDataAsZero --binSize 10 --afterRegionStartLength 15000 --beforeRegionStartLength 15000`), followed by the *plotHeatmap* function. *petMar3* gene predictions were downloaded (J. J. Smith *et al.* 2018) and overlap with eliminated sequences was identified using *bedtools* *intersect* function.

We used the `-batch` option in CNVkit to identify deletions, which runs the CNVkit pipeline on samples. Please note that a target file is not required for CNVkit if `-m wgs` is selected. CNVkit will output a final report of absolute copy number gains and losses, which we used to identify eliminated sequences in blood. Given the high degree of overlap between regions identified using CNVkit and previously published eliminated sequences (**Figure 4A**), many of which were experimentally validated through PCR assays by Smith *et al.* 2018, the default settings of CNVkit were sufficient for our analysis. We have also included a citation for CNVkit, which describes the function of the tool in greater detail. As seen in **Figure 4C** and **Supplementary Figure S5D-E** there is clear germline (sperm) read enrichment and somatic read depletion across three independent replicates (Smith *et al.* data and two biological replicates sampled for this study) and two distinct sequencing techniques (Nanopore long read and Illumina short read technologies).

We would also like to clarify that we identified eliminated sequences using previously published short-read whole-genome sequencing data from Smith *et al.*, and not WGBS data that we generated for this study. We have now merged the methods sections “Whole-genome sequencing read assembly” and “Identification of sequences eliminated during PGR” to clarify which data was used for this purpose. In addition, our subsequent analyses are not based on the CNVkit output alone; all analyses are performed using regions that were identified by CNVkit that overlap eliminated sequences which have been previously identified by Smith *et al.* (many of which were experimentally validated using PCR assays by Smith *et al.* and are cross-referenced by Timoshevskaya *et al.* in confirmation of eliminated sequences in the kPetMar1 genome). This was done to ensure that our findings were based on a stringent dataset that was generated using multiple independent methods.

A quick comparison of the results of the current paper vs the results in @timoshevskaya2023, indicates that Angeloni *et al.* (2023) identified 112 chromosomes/scaffolds covering 5.23 Mb that were eliminated from somatic genome. In contrast, Timoshevskaya *et al.* (2023) identified 29.119 germline-enriched Mb across 315 chromosomes/scaffolds. The overlap between the two

approaches encompassed 4.937 Mb across 101 regions. The assembled eliminated chromosomes from Angeloni et al. (2023) included parts of chromosomes 3, 6, 13, 18, 30, 35, 39, 44, 48, 81. Conversely, Timoshevskaya et al. (2023) identified only chromosomes 51, 81 as well as many scaffolds. Most of the regions identified by CNVkit covered less than 50% of a scaffold/chromosome sequence, whereas DiffCover found that the majority or entirety of a chromosome or scaffold was removed and only three regions had <50% of eliminated sequence: chr 51, NW_022638341.1, NW_022639355.1 (see figure below). Since chromosomal diminution refers to the shortening of chromosomes after programmed genome rearrangement (PGR), while chromosomal elimination involves removal of entire chromosomes, the results imply different processes of PGR. Since there is other evidence for chromosomal elimination (other papers by the Smith lab) not chromosome diminution, and the results of the Angeloni study are based on WGBS, the results of the Angeloni paper likely identified regions that are differentially methylated not eliminated *per se*. In any case, it would make sense to revise the manuscript accordingly, so as not to introduce confusion into this unresolved issue.

RESPONSE 3: We thank the Reviewer for taking the time to analyze our data, and we certainly agree that it is an important point to resolve before our manuscript can be considered for publication. We have now performed additional analysis of the regions we identified as eliminated using CNVkit, and those identified as eliminated by Smith *et al*, using the petMar3 reference genome. As seen in **Review Figure 1A**, the size distributions of sequences published by Smith *et al* and those identified by CNVkit in our study are highly similar. We also found that sequences identified by CNVkit and those published by Smith *et al* mostly cover > 75% of entire scaffolds in the petMar3 genome (**Review Figure 1B**), and that there is considerable overlap between scaffolds that are covered at least 50% by eliminated sequences identified in each method (**Review Figure 1C**). This is in line with reports of chromosome elimination in lamprey. Accordingly, we did not suggest or discuss the possibility that chromosome diminution is occurring during PGR in lamprey. We also studied 5mC levels at the eliminated sequences in the Smith *et al* study, where eliminated sequences largely represent entire scaffolds. We found that these sequences are hypermethylated (**Review Figure 1D**); thus, our conclusion does not change whether we consider the entire scaffold or the more stringent dataset that includes only regions identified by both CNVkit and Smith *et al*. Finally, and as discussed in more detail above, we did not use DNA methylation (WGBS) data for the identification of eliminated sequences, thus our results cannot be confounded by potential differences in DNA methylation.

Review Figure 1. A) Length distribution of eliminated sequences identified by CNVkit using the petMar3 reference genome, and those published by Smith *et al*. Dotted line represents median length for each dataset. **B)** Histograms depicting the percentage of entire scaffolds in the petMar3 reference genome that are covered by eliminated sequences identified using CNVkit and those published by Smith *et al*. **C)** Venn diagram of the overlap between scaffolds that are covered lengthwise at least 50% by Smith *et al* eliminated sequences, and scaffolds that are covered lengthwise at least 50% by CNVkit eliminated sequences. **D)** Heatmaps depicting mCG signal from Nanopore and WGBS sperm samples in two biological replicates at eliminated sequences published by Smith *et al*.

We then performed a liftover of the previously published sequences and the ones identified by CNVkit to the kPetMar1 genome using tools from the kentUtils suite (<https://github.com/ENCODE-DCC/kentUtils>). The reference genome was split into 3kb bins and converted to lift files using faSplit option. psl files were generated using blat function (-t=dna -q=dna -tileSize=12 -fastMap -minIdentity=95 -noHead -minScore=100) and were converted to parent coordinate system using liftUp function (-pslQ) followed by generation of chain files using axtChain function (-linearGap=medium -faQ -faT). Resulting chain files were merged and sorted (chainMergeSort *.chain | chainSplit chain_split stdin), alignment nets were generated (chainNet) and the liftOver chain file was produced (netChainSubset). Liftover of eliminated sequences from petMar3 to kPetMar1 was performed using the generated chain file (liftOver -minMatch=0.45).

When we compared the lifted eliminated regions to those identified by Timoshevskaya *et al*, we found that sequences from the Smith *et al* paper ($n = 311$, 7.09Mbp) and those identified by CNVkit ($n = 342$, 5.76Mbp) were largely similar to each other, but distinct from those identified by Timoshevskaya *et al* ($n = 357$, 29.12Mbp), as was found by the Reviewer. As seen in **Review Figure 2A**, the length distribution of the CNVkit and Smith *et al* liftover eliminated sequences are much shorter than those identified by Timoshevskaya *et al*, and they cover a much smaller percentage of the scaffolds that they overlap (**Review Figure 2B**).

Review Figure 2. A) Length distribution of eliminated sequences identified by CNVkit and those published by Smith *et al*, both lifted from petMar3 to kPetMar1, and those identified by Timoshevskaya *et al* using the kPetMar1 reference genome. Dotted line represents median length for each dataset. **B)** Histograms depicting the percentage of entire scaffolds in the kPetMar1 reference genome that are covered by eliminated sequences identified using CNVkit and those published by Smith *et al* lifted to kPetMar1, and those identified by Timoshevskaya *et al*.

While we certainly think that the Reviewer’s initial suggestion of lifting these coordinates to the latest genome release was a good idea, after some discussion we do not think that liftover is an optimal approach for comparison to the new genome reference for several reasons. First, these sequences mostly represent entire scaffolds, and the UCSC liftOver tool does not perform as well with large genomic domains (see: <https://genome.ucsc.edu/goldenPath/help/liftOver.html> - “It should also be noted that tracks containing large regions will not lift as well because of the increased chance of spanning a region that has changed between the two assemblies.”). Secondly, eliminated sequences contain highly repetitive sequences, which poses challenges when accurately assigning their location in the new reference genome. We would also like to point out that although the kPetMar1 reference is an improved assembly of the germline genome, the petMar3 reference genome is a high quality assembly, with comparable assembly

lengths (1.13Gbp for petMar3 v 1.09Gbp for kPetMar1) and scaffold N50 (11.93Mbp v 13.00Mbp) and L50 (34 v 33) values. Please note that the kPetMar1 genome was under embargo until recently, so we were working with what was available to us at the time of analysis. It is also important to consider that we cannot rule out the possibility that there are differences between individuals or populations in the sequences that are eliminated, which would also impact the liftover results if true; this is clearly stated by Timoshevskaya *et al* in their paper.

In sum, we have now updated our manuscript to remove the supplementary table containing the sequences that were lifted over as well as any reference to this table in the manuscript, and have removed the methods section which details how the liftover was performed.

REVIEWERS' COMMENTS

Reviewer #1 (Remarks to the Author):

Thanks for revisiting this and for the nice contribution to understanding changes in methylation during PGR in sea lamprey. It makes sense to me that using the lift-over tool would not work well since there is a fairly large discrepancy between the 2018 version (hosted at SIMR) and the newest version of the genome hosted at NCBI. I understand why the authors don't won't to redo the analyses with respect to the newer genome - hopefully in a future work this can be done (at which point, the reference genome may also be further improved).